# Gαi2-induced conductin/axin2 condensates inhibit Wnt/β-catenin signaling and suppress cancer growth

Cezanne Miete [1], Gonzalo P. Solis [2], Alexey Koval[2], Martina Brückner[1], Vladimir L. Katanaev [2,3], Jürgen Behrens[1] & Dominic B. Bernkopf [1✉]

Conductin/axin2 is a scaffold protein negatively regulating the pro-proliferative Wnt/β-catenin signaling pathway. Accumulation of scaffold proteins in condensates frequently increases their activity, but whether condensation contributes to Wnt pathway inhibition by conductin remains unclear. Here, we show that the Gαi2 subunit of trimeric G-proteins induces conductin condensation by targeting a polymerization-inhibiting aggregon in its RGS domain, thereby promoting conductin-mediated β-catenin degradation. Consistently, transient Gαi2 expression inhibited, whereas knockdown activated Wnt signaling via conductin. Colorectal cancers appear to evade Gαi2-induced Wnt pathway suppression by decreased Gαi2 expression and inactivating mutations, associated with shorter patient survival. Notably, the Gαi2-activating drug guanabenz inhibited Wnt signaling via conductin, consequently reducing colorectal cancer growth in vitro and in mouse models. In summary, we demonstrate Wnt pathway inhibition via Gαi2-triggered conductin condensation, suggesting a tumor suppressor function for Gαi2 in colorectal cancer, and pointing to the FDA-approved drug guanabenz for targeted cancer therapy.

[1] Experimental Medicine II, Nikolaus-Fiebiger-Center, Friedrich-Alexander University Erlangen-Nürnberg, 91054 Erlangen, Germany. [2] Department of Cell Physiology and Metabolism, Centre Médical Universitaire, University of Geneva, 1211Geneva 4, Geneva, Switzerland. [3] School of Biomedicine, Far Eastern Federal University, 690922 Vladivostok, Russia. ✉email: dominic.bernkopf@fau.de

The Wnt/β-catenin signaling pathway tightly controls the expression levels of the mitogenic transcriptional co-factor β-catenin, and deregulation of the pathway is causally associated with diverse human pathologies[1]. Most prominently, aberrantly activated Wnt signaling drives carcinogenesis in more than 90% of colorectal cancers[2]. The cytoplasmic proteins axin and its homolog conductin/axin2 are negative regulators of the Wnt pathway acting as scaffold proteins of the β-catenin destruction complex to promote β-catenin degradation[3,4]. In contrast to axin, which is constitutively expressed, conductin expression is initiated by β-catenin[5,6]. Thus, conductin functions as a negative feedback regulator to limit pathway activity. As a β-catenin target gene, conductin is highly expressed in colorectal cancer, but despite upregulation fails to suppress Wnt signaling and to prevent cancer growth[5]. We recently showed that conductin´s activity is partially suppressed by a short aggregating protein sequence in its regulator of G-protein signaling (RGS) domain[7]. RGS–RGS aggregation mediated by this aggregon prevents polymerization of conductin via its DIX domain, which is essential for efficient β-catenin degradation[7,8]. The aggregon is not conserved in axin, which accordingly shows efficient DIX-mediated polymerization and β-catenin degradation. Inactivation of conductin RGS aggregation either by aggregon mutation or a synthetic aggregon-targeting peptide unleashes conductin´s full activity leading to enhanced polymerization and β-catenin degradation[7].

Axin polymers, which are visible as spherical cytoplasmic structures and therefore called puncta, were recently characterized as biomolecular condensates[9,10]. Biomolecular condensates are membrane-less organelles forming via liquid-liquid phase separation of macromolecules[11,12]. They are thought to act as super scaffolds by concentrating or excluding effector proteins and may play pivotal roles in signal transduction[13–15]. Although conductin is rather diffusely distributed, its polymers obtained after blocking the aggregon form likewise puncta and can be considered as condensates as well. The RGS aggregon might therefore represent a signaling hub by which cellular interactors of the RGS domain regulate conductin condensation for physiological Wnt pathway control.

There is evidence that trimeric G-proteins modulate Wnt signaling activity. Direct interactions of Gα proteins or Gβγ-subunits with Wnt pathway components, such as Gαo-axin, Gαs-axin, Gα13-disheveled, or Gβγ-disheveled binding have been described to activate Wnt signaling[16–19]. Gα12 and Gα13 are able to release β-catenin from cell junctions that is otherwise not available for signaling, thereby enhancing Wnt pathway activity in cancer cells with impaired β-catenin degradation[20,21]. Indirectly, Gαs activates Wnt signaling by promoting protein kinase A (PKA) activity, which phosphorylates β-catenin to increase its stability and transcriptional activity[22–25]. In contrast to activation, Wnt pathway inhibition by G-proteins remains poorly studied. As Gα proteins were shown to interact with the axin RGS domain[16,17,26], they are good candidates for targeting the conductin RGS aggregon.

Here, we show that the Gα protein i2 (Gαi2) interacts with the conductin RGS domain and interferes with aggregation by masking the aggregon, thereby promoting conductin condensation and β-catenin degradation. Consistently, we characterize Gαi2 in knockdown and transient expression experiments as potent negative Wnt pathway regulator acting via conductin. Our findings have implications for colorectal cancer development and treatment: human cancer data suggest silencing of a Gαi2 tumor suppressor function during colorectal carcinogenesis. Moreover, the Gαi2-activating drug guanabenz (GBZ) inhibits the growth of Wnt signaling-driven colorectal cancer in vitro and in vivo suggesting that aggregon targeting can be exploited for therapy.

## Results

**The RGS interactor Gαi2 induces conductin condensation.** To identify proteins that regulate Wnt signaling by targeting the conductin aggregon, we analyzed the binding mode of known RGS interactors in structural models. For adenomatous polyposis coli (APC), an RGS binding Wnt pathway component, structural modeling suggests no interference of binding with RGS aggregation, as both sites are clearly distinct (Supplementary Fig. 1a, b). Thus, we focused on Gα subunits of trimeric G-proteins, the most prominent interactors of RGS proteins. Here, structural models based on published X-ray data of Gαi/RGS complexes superimposed on the conductin RGS domain predict binding in close proximity to the aggregon, suggesting competition with RGS aggregation (Fig. 1a and Supplementary Fig. 1c, d). Indeed, transient expression of Gαi2 changed the cellular distribution of co-expressed conductin from a diffuse pattern to spherical polymers, i.e., condensates (Fig. 1b, c), indicative of reduced RGS aggregation[7]. Out of four investigated Gα proteins of the Gαi/o family, markedly Gαi2 and to some degree Gαi3 induced condensation, while highly related Gαi1 and Gαo were much less efficient, independent of expression levels (Fig. 1c–f). In addition to condensation, Gαi2 occasionally recruited conductin to the plasma membrane (Fig. 1c arrowheads, e). Importantly, a DIX domain-inactive conductin mutant (CdtM3) showed only membrane recruitment but no condensation indicating that Gαi2-induced condensates are formed via DIX domain-mediated polymerization (Fig. 1b–e), a key indicator for specific assembly of axin proteins[8,27]. Gαi2-induced condensation of conductin was consistently observed independent of the cell line (Supplementary Fig. 2a–e), protein-tagging (Supplementary Fig. 2f–h) and the immunofluorescence fixation method (Supplementary Fig. 2i–k). Investigating other Gα protein families revealed that Gαs and Gαq were rather ineffective while Gα12 triggered conductin polymerization (Supplementary Fig. 3a–c).

**Gαi2 induces conductin condensation via aggregon-targeting.** Binding of Gαi2 to the conductin RGS domain was indicated by co-localization of both proteins in condensates (insets Fig. 1c and Supplementary Fig. 2b, f, i) and confirmed in pulldown assays using the isolated RGS domain (Fig. 2a, b). Moreover, RGS domain binding of different Gα proteins perfectly correlated with condensation ($R = 0.99$, $p = 0.011$), suggesting a functional connection between G-protein binding and polymerization (Supplementary Fig. 4a). Consistently, a G-protein mutation that is associated with impaired RGS protein interaction (G184S)[28,29] reduced binding of Gαi2 to the conductin RGS domain (Fig. 2c and Supplementary Fig. 4b), and attenuated conductin polymerization and membrane recruitment (Fig. 2d, e). Subcategorization of cells with conductin condensates revealed that particularly cells with a high number of condensates were observed less frequently upon expression of the G184S mutant compared to WT (Supplementary Fig. 4c, d). Studies with purified recombinant proteins showed that the interaction of the conductin RGS domain with inactive GDP-loaded Gαi2 was significantly increased upon Gαi2 activation either via AlF$_4^-$ treatment or via loading with the hydrolysis-resistant GTP analog GTPγS (Fig. 2f and Supplementary Fig. 4e). Aluminum fluoride binds to the GDP pocket of GDP-loaded Gα proteins thereby mimicking the GTP-loaded state[30]. Despite preferential binding to active Gαi2, we did not observe any GTPase activating protein (GAP) function of the conductin RGS domain (Supplementary Fig. 4f, g). In line with the pulldown experiments (Fig. 2f), treatment of cells with AlF$_4^-$ enhanced induction of conductin condensation by transiently expressed Gαi2 (Supplementary Fig. 5a) and, notably, even without Gαi2 expression (Supplementary Fig. 5b, c). Importantly, AlF$_4^-$-induced condensation was markedly reduced upon

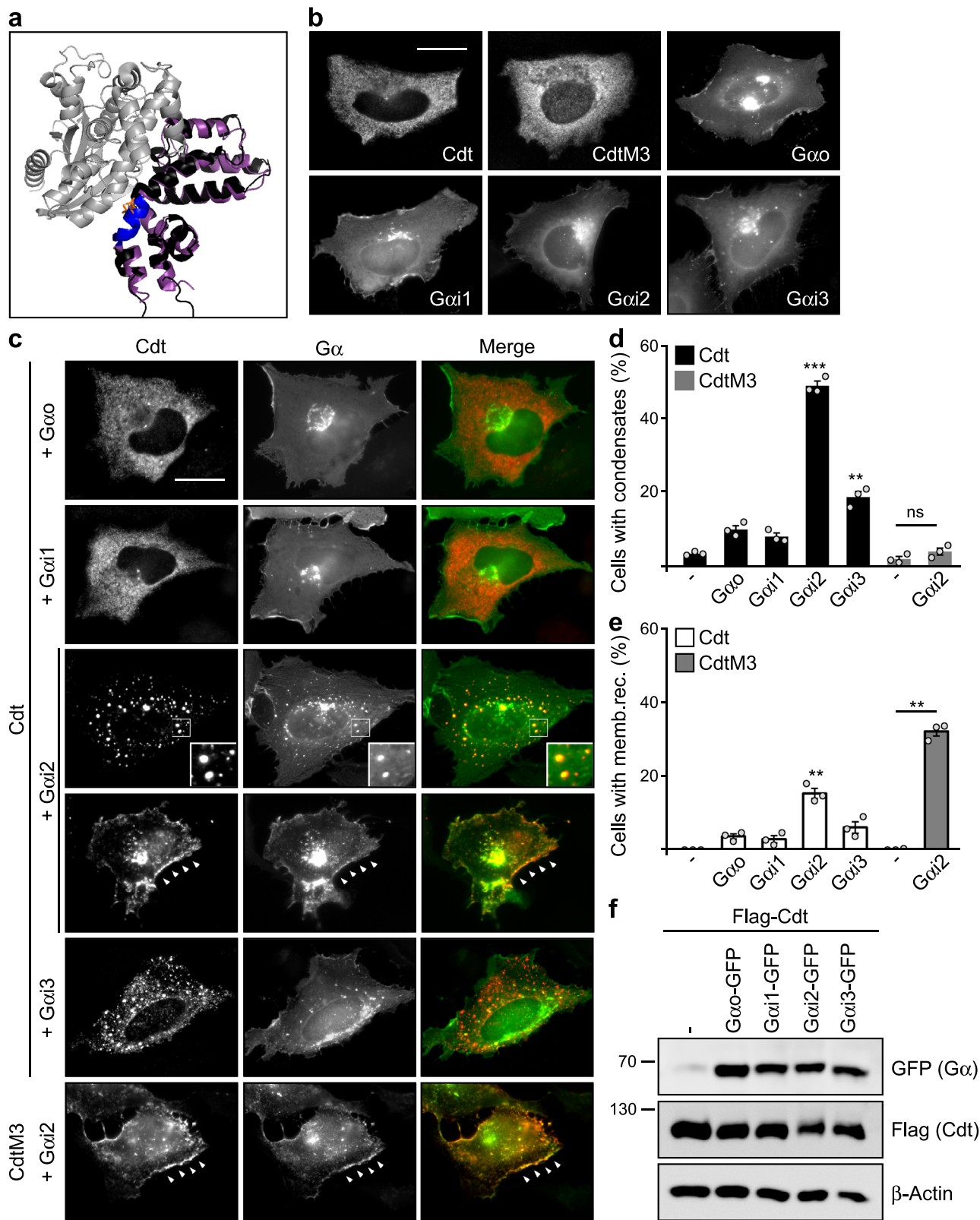

stable knockdown of Gαi2 or Gαi3 but not of Gαi1 compared to control cells, indicating that binding of endogenous Gαi2 (and 3) initiates conductin polymerization (Fig. 2g, h and Supplementary Fig. 5d). Gα binding may interfere with RGS aggregation given the close proximity of the binding site to the aggregon (Fig. 1a). To test this, Gαi proteins were co-expressed with a conductin fragment (Cdt 2–345) forming RGS-RGS aggregates, which can be detected

as high molecular weight complexes by native gel electrophoresis[7]. Of note, co-expression of Gαi2 decreased these high molecular weight complexes and increased the amount of smaller complexes indicative for reduced RGS aggregation (Fig. 2i, j and Supplementary Fig. 5e). Likewise, in line with promoting conductin polymerization (Fig. 2h), AlF$_4^-$ treatment decreased RGS aggregation (Fig. 2k).

**Fig. 1 Gαi2 induces conductin condensates. a** Structural alignment of the modeled conductin RGS domain (black) to RGS1 (purple) co-crystalized with Gαi1 (gray) (PDB ID: 2GTP)[69]. The conductin aggregon and its two key residues (QV) are highlighted in blue and orange, respectively. **b, c** Immunofluorescence staining of Flag (red) in U2OS cells transfected with Flag-tagged conductin (Cdt), its M3 mutant, and different GFP-tagged Gα proteins (green) either alone (**b**) or in combinations (**c**), as indicated. For co-expression of conductin with Gαi2, examples of cells with condensate formation (upper row) and membrane recruitment (lower row) are shown. Insets are magnified in the lower right corner. Arrowheads point to membrane recruitment. Scale bars: 20 μm. **d, e** Percentage of 900 transfected cells out of three independent experiments as in (**c**) that show condensation (**d**) or membrane recruitment (**e**) of conductin or the M3 mutant. Results are mean ± SEM (*n* = 3). **\**p* < 0.01, \**\**p* < 0.001 (two-sided Student's *t*-test). **f** Western blotting for GFP, Flag, and β-actin (loading control) in lysates of U2OS cells transfected with indicated constructs. One out of three representative experiments is shown. Molecular weight is indicated in kDa. Source data and exact *p*-values are provided as source data file.

We proposed that Gαi2 induces conductin condensation via steric competition with aggregation, while Gαi2 binding without affecting aggregation will not suffice (Fig. 3a). To probe the hypothesis, we generated a conductin mutant with a novel aggregation site (red) that should not be accessible to Gαi2 binding (Fig. 3b). To this end, we inactivated the functional aggregon (blue) by Q188P and V189S substitutions and activated an otherwise silent aggregation site (red) by L99R substitution (Supplementary Fig. 6a), using previously identified mutations[7,31]. Inactivation of the blue aggregon (QV mutant) resulted in spontaneous condensation (Supplementary Fig. 6b, c) and reduced RGS aggregation, detected by low-weight aggregation complexes in native gels (Supplementary Fig. 6d) and less efficient penetration in fractions of high sucrose density upon ultracentrifugation (Supplementary Fig. 6e), as we described previously[7]. Additional activation of the red aggregation site (QVL mutant) completely rescued diffuse cellular distribution, the high molecular weight aggregation complexes in native gels, and penetration in fractions of high sucrose density (Supplementary Fig. 6b–e). Thus, RGS aggregation via the red aggregation site fully compensates for inactivation of the blue aggregon. Importantly, co-expression of Gαi2 failed to induce condensation of conductin QVL (Fig. 3c, d). In contrast, membrane recruitment was still intact, indicating that the QVL mutation does not decrease Gαi2–RGS interaction (Fig. 3c, e), which was confirmed in pull-down assays (Fig. 3f, g). In fact, the QVL mutated RGS domain showed stronger Gαi2 binding compared to WT, possibly because of competition between Gαi2 binding and RGS aggregation in WT conductin but not in the QVL mutant (Fig. 3f, g). Together, our experiments show that mere binding of Gαi2 is not sufficient to promote condensation, and support that Gαi2 induces conductin condensation by decreasing RGS aggregation (Fig. 3a).

**Conductin polymerizes independently of Gαi2-PKA signaling.** Gαi2 signaling leads to a decrease of cAMP levels, followed by loss of PKA activity, and reduced transcription of target genes with cAMP-responsive elements (CRE)[32]. We, therefore, analyzed whether cAMP/PKA signaling might be involved in Gαi2-induced conductin condensation. However, direct PKA inhibition by the chemical inhibitor H-89 failed to induce conductin condensates or membrane recruitment (Supplementary Fig. 7a–d), although we confirmed equivalent PKA inhibition by Gαi2 and H-89 using a CRE-luciferase reporter (Supplementary Fig. 9b, d). Furthermore, rescuing the Gαi2-induced decrease of cAMP levels either via the adenylyl cyclase activator forskolin or via a cell-permeable cAMP analog (Bt₂cAMP) did not impair condensation (Supplementary Fig. 7e, g, j), in spite of efficient rescue as shown by the CRE-luciferase reporter and a red fluorescent indicator for cAMP (R-FlincA)[33] (Supplementary Fig. 7f, h, i, k–n). These data indicate that Gαi2 signaling via cAMP/PKA seems to be irrelevant for promoting polymerization of conductin.

**Gαi2 inhibits Wnt signaling by promoting conductin activity.** We next analyzed whether Gαi2-induced conductin condensation affects Wnt/β-catenin signaling. SW480 colorectal cancer cells exhibit high endogenous β-catenin levels, which are reduced by transient conductin expression (Fig. 4a). Importantly, Gαi2 co-expression enhanced conductin-mediated β-catenin degradation, and this was further amplified by additional AlF₄⁻ treatment, rendering WT conductin as active as the aggregation-dead, spontaneously polymerizing QV mutant (Fig. 4a, b and Supplementary Fig. 6b). In contrast, Gαi2 and AlF₄⁻ did not enhance β-catenin degradation by the conductin mutants QVL and M3, which were both incapable of forming condensates upon Gαi2 co-expression (Figs. 1d and 3c–e), suggesting specific Gαi2 function via induction of condensation (Fig. 4a, b and Supplementary Fig. 8a–c). In addition, Gαi2 enhanced the degradation of transiently expressed β-catenin by co-expressed conductin in HEK293T cells in a dosage-dependent manner (Fig. 4c). Consistently, knockdown of Gαi2 by two independent siRNAs strongly activated Wnt/β-catenin signaling in different cell lines (Fig. 4d, e and Supplementary Fig. 8d), as measured by the TOP-Flash reporter and conductin amounts. While conductin knockdown activated the Wnt pathway, as expected, no further activation was observed upon additional Gαi2 knockdown (Fig. 4f and Supplementary Fig. 8e), suggesting that Gαi2-induced pathway repression requires conductin. As a control, Wnt treatment further activated the pathway in conductin knockdown cells (Supplementary Fig. 8f, g). In line with the knockdown data, Gαi2 expression inhibited in a dosage-dependent manner both the Wnt3a-induced signaling in cells with otherwise low pathway activity (Fig. 4g, h and Supplementary Fig. 8h–j) and the basal Wnt signaling in colorectal cancer cell lines with high endogenous pathway activity (Fig. 4i, k). The Gαi2 G184S mutant inhibited Wnt signaling less efficiently (Supplementary Fig. 8i, j), consistent with reduced binding to the conductin RGS domain (Fig. 2c). Wnt pathway inhibition by Gαi2 depended at least partially on conductin, as it was markedly attenuated by knockdown or CRISPR/Cas9-mediated knockout of conductin (Fig. 4h–l and Supplementary Fig. 8k). The residual Gαi2 activity in conductin knockout cells (Fig. 4k) most likely depends on PKA inhibition, a kinase activating Wnt signaling via β-catenin phosphorylation[22]. However, side-by-side comparisons revealed about twofold stronger Wnt pathway inhibition by Gαi2 compared to specific PKA inhibition by H-89 (Supplementary Fig. 9a, c), although PKA inhibition by Gαi2 and H-89 was similar as verified via a CRE-luciferase reporter (Supplementary Fig. 9b, d). This indicates inhibition of Wnt signaling by Gαi2 via another mechanism in addition to PKA inhibition, such as promoting conductin activity. In contrast, inhibition of Wnt signaling via PKA inhibition appeared to be conductin independent, as H-89 inhibited the pathway to a similar extent in conductin WT and knockout cells (Supplementary Fig. 9e). Together, our data suggest that Gαi2 negatively regulates Wnt/β-catenin signaling in a variety of different cell lines including colorectal cancer cells by promoting conductin polymerization and by conductin-independent inhibition of PKA.

**Gαi2 aberrations are associated with reduced patient survival.** Since Gαi2 inhibited Wnt signaling in colorectal cancer cells, we wondered whether Gαi2 suppresses colorectal carcinogenesis. Analysis of mRNA expression data revealed a highly significant negative

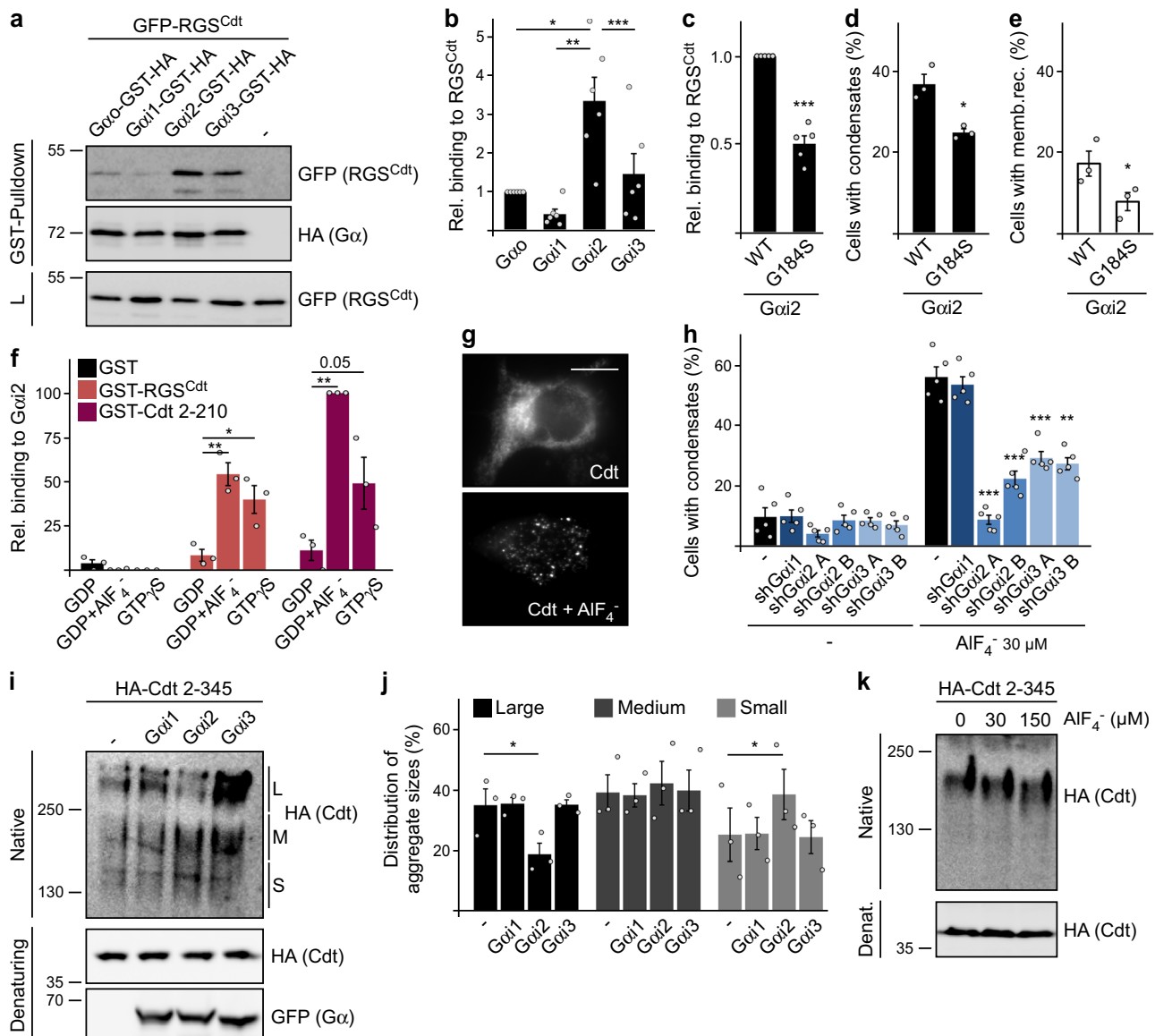

**Fig. 2 Gαi2 induces conductin polymerization via steric interference with RGS–RGS aggregation. a** Western blotting for GFP and HA in lysates (L) of HEK293T cells transfected with indicated constructs, and after GST-pulldown from these lysates. **b** Quantification of the RGS domain amounts that were co-precipitated with the indicated Gα proteins in six independent experiments as in (**a**). RGS amounts are normalized to the precipitated amounts of the Gα proteins and presented relative to Gαo. **c** Quantification of RGS domain binding as described in (**b**) from five independent experiments as shown in Supplementary Fig. 4b. **d, e** Percentage of 900 transfected cells out of three independent experiments similarly performed as in Fig. 1c that show condensation (**d**) or membrane recruitment (**e**) of conductin. **f** Relative binding of purified GST (control) and GST-tagged conductin fragments containing the RGS domain (GST-RGSCdt, GST-Cdt 2–210) to purified Gαi2 preloaded with GDP, GDP + AlF4⁻, or GTPγS, quantified from three independent pulldown experiments with purified recombinant proteins, as shown in Supplementary Fig. 4e. **g** Immunofluorescence staining of HA-Conductin (Cdt) in HEK293 shRNA-control cells which were untreated or treated with 30 µM AlF4⁻ for 1 h. Scale bar: 10 µm. **h** Percentage of 1500 transfected cells out of five independent experiments as in (**g**) that show condensation of HA-Conductin. The cells stably express either a control shRNA (−) or indicated shRNAs against Gα proteins. **i, k** Western blotting under native and denaturing (denat.) conditions for HA and GFP in lysates of HEK293T cells, which were transfected and treated with AlF4⁻, as indicated. **j** Western blot quantification of large, medium, and small aggregation complexes according to indications L, M, and S in (**i**) as percentage of the total protein amount, from three independent experiments. Results are mean ± SEM (n = 6 [**b**], n = 5 [**c, h**], n = 3 [**d–f, j**]). *p < 0.05, **p < 0.01, ***p < 0.001 (two-sided Student's t-test). Molecular weight is indicated in kDa (**a, i, k**). Source data and exact p-values are provided as source data file.

correlation between expression levels of *GNAI2* (encoding Gαi2) and β-catenin target genes in human colon tissue (Fig. 5a and Supplementary Fig. 10), in line with Wnt signaling inhibition by Gαi2 in vivo. In colorectal cancer, The Cancer Genome Atlas (TCGA) data sets COAD and READ showed loss of *GNAI2* gene copies in about 5% of the cases (Fig. 5b), and decreased *GNAI2* mRNA expression compared to healthy tissue (Fig. 5c). The negative correlation

between expression of *GNAI2* and β-catenin target genes was deteriorated in cancer, as expected considering the reduced *GNAI2* expression, however it was not abolished (Fig. 5a and Supplementary Fig. 10). Consistent with the remaining negative correlation, siRNA-mediated Gαi2 knockdown activated Wnt signaling in DLD1 and SW480 colorectal cancer cells, and this was strongly attenuated in SW480 conductin knockout cells (Fig. 5d, e). As knockdown of

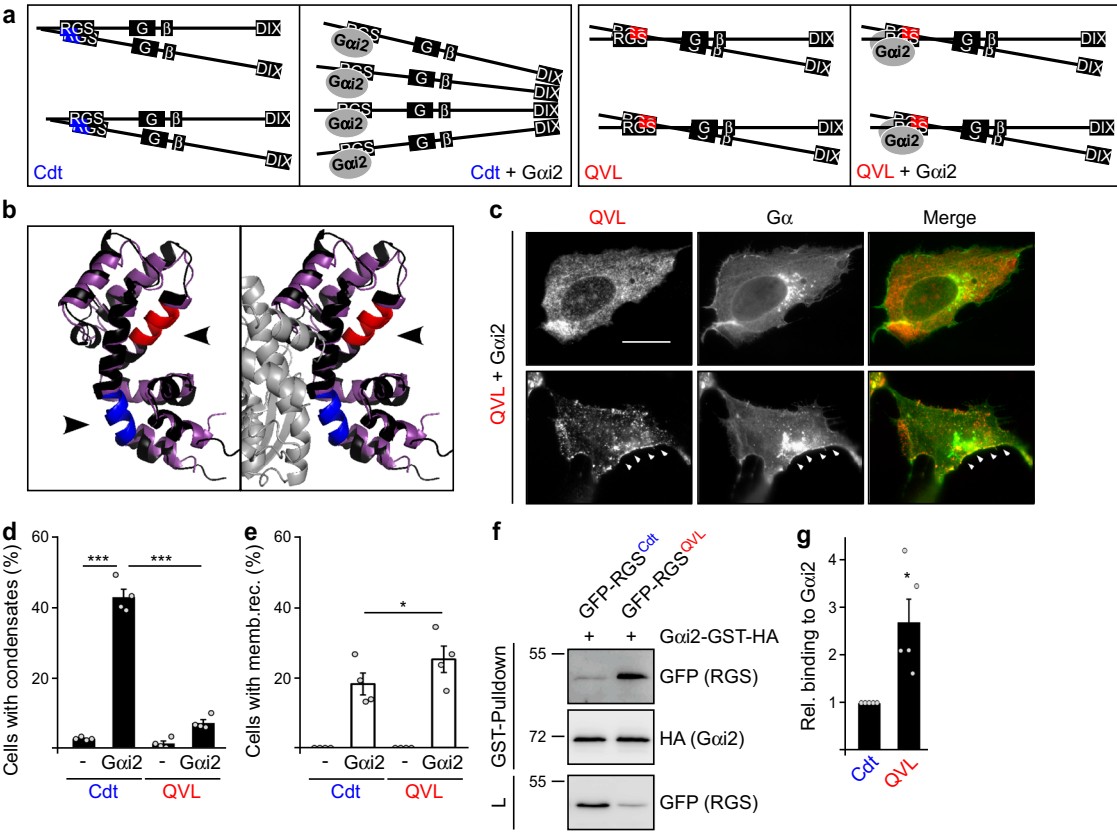

**Fig. 3 Induction of conductin polymerization requires aggregon masking. a** Schematic presentation showing DIX domain-mediated conductin polymerization after RGS aggregation mediated by the blue aggregon gets sterically inhibited by Gαi2 binding (left), contrasting the persisting RGS aggregation mediated by the red aggregation site in spite of Gαi2 binding in the QVL mutant (right). GSK3 (G) and β-catenin (β) binding sites are indicated. **b** Structural alignment of the modeled conductin RGS domain (black) to the structure of RGS1 (purple) co-crystalized with Gαi1 (gray) (PDB ID: 2GTP). The conductin aggregon and the silent aggregation site are highlighted in blue and red, respectively. Arrowheads indicate the accessibility of the aggregation sites without and with Gα protein binding. **c** Immunofluorescence staining for Flag (red) in U2OS cells transfected with Flag-Conductin QVL together with Gαi2-GFP (green). Arrowheads point to membrane recruitment. Scale bar: 20 μm. **d, e** Percentage of 1200 transfected cells out of four independent experiments as in (**c**) that show condensation (**d**) or membrane recruitment (**e**) of conductin or the QVL mutant. **f** Western blotting for GFP and HA in lysates (L) of HEK293T cells transfected with indicated constructs, and after GST-pulldown from these lysates. Molecular weight is indicated in kDa. **g** Quantification of Gαi2-binding to the WT and QVL-mutated conductin RGS domain from five independent GST-pulldown experiments as in (**f**). Results are mean ± SEM ($n = 4$ [**d, e**], $n = 5$ [**g**]). *$p < 0.05$, ***$p < 0.001$ (two-sided Student's $t$-test). Source data and exact $p$-values are provided as source data file.

endogenous Gαi2 activated Wnt signaling in colorectal cancer cells, the observed decrease of Gαi2 expression in human tumors may likewise allow higher Wnt pathway activity, which promotes carcinogenesis. In addition to reduced expression, Gαi2 missense mutations occur in colorectal cancer in about 1% of the cases (COAD: 5/400; READ: 1/137). Of note, such Gαi2 cancer mutations markedly impaired Wnt signaling inhibition independent of expression levels (Fig. 5f–h and Supplementary Fig. 11a, b). In line with our proposed mechanism, the cancer mutations attenuated Gαi2-conductin interaction (Supplementary Fig. 11c, d) and induction of conductin polymerization (Supplementary Fig. 11e). As observed for the attenuated G184S mutant before (Supplementary Fig. 4c, d), in particular cells with a high number of condensates occurred less frequently upon expression of the cancer mutants compared to WT (Supplementary Fig. 11f, g). Finally, survival of patients with *GNAI2* copy number loss or missense mutations (about 6% of the cases) was significantly reduced: more than 50% of the patients with *GNAI2* alterations are estimated to die within about three years after diagnosis compared to 25% of the patients without (Fig. 5i). Collectively, our data indicate evasion of colorectal cancers from Gαi2-mediated Wnt pathway suppression via decreased expression and/or inactivating mutations. We thereby suggest a role for Gαi2 as a tumor suppressor whose inactivation is associated with reduced patient survival.

**Activating endogenous Gαi2 inhibits Wnt signaling via conductin.** Next, we asked whether activation of endogenous Gαi2 would lead to inhibition of Wnt/β-catenin signaling. Guanabenz (GBZ) is a specific FDA-approved agonist of α2-adrenoceptors, which are G-protein-coupled receptors (GPCR) activating Gαi signaling[34]. Importantly, GBZ treatment induced condensation of transiently expressed conductin (Fig. 6a, b and Supplementary Fig. 12a), and this was abolished in Gαi2 knockdown cells showing dependency on Gαi2 (Supplementary Fig. 12b). Moreover, time-lapse microscopy revealed frequent initiation of GBZ-induced condensation in close proximity to plasma membranes, consistent with GPCR-mediated Gαi2 activation (Supplementary Fig. 12c and Supplementary Movie 1). In addition, we observed fusion of conductin puncta (Supplementary Fig. 12d), initiation of polymerization at areas of high concentration (Supplementary Fig. 12e), and enhanced polymerization at lower temperatures (Supplementary Fig. 12f), which are all typical features of biomolecular condensates[12]. In line with our proposed mechanism

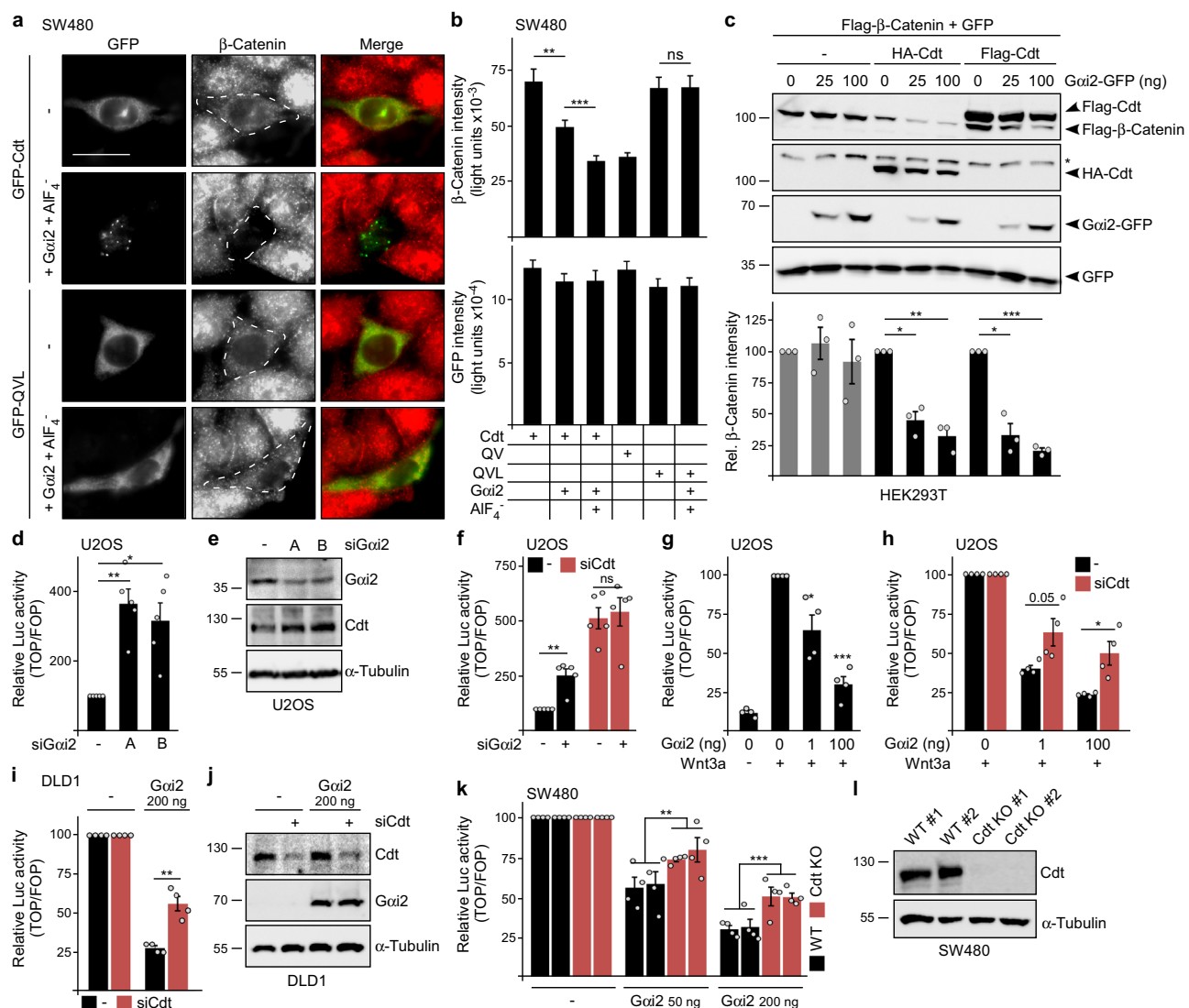

**Fig. 4 Gαi2 promotes conductin-mediated inhibition of Wnt signaling. a** Immunofluorescence staining of endogenous β-catenin (red) in SW480 cells, which were transfected and treated as indicated on the left. Scale bar: 20 μm. **b** Quantification of β-catenin and GFP fluorescence intensity in four independent experiments as in (**a**). **c** Western blotting for Flag, HA, and GFP in lysates of HEK293T cells transfected as indicated above the blots. GFP: loading and transfection control. * indicates unspecific band. Quantification of Flag-β-catenin normalized to GFP of three independent experiments. **d, f–h** Luciferase activity (TOP/FOP) in U2OS cells transfected with indicated siRNAs (**d, f**), or/and indicated amounts of Gαi2 (**g, h**). Wnt3a treatment is indicated. **e** Western blotting showing the efficiency of Gαi2 knockdown in U2OS cells. Consistent with TOP-Flash activation (**d**), expression of the β-catenin target gene conductin (Cdt) was increased upon knockdown. α-Tubulin: loading control. **i, k** Luciferase activity (TOP/FOP) in Gαi2 transfected DLD1 cells without (−) and with conductin knockdown (**i**), and in parental SW480 cells, a WT control clone and two CRISPR/Cas9 *AXIN2/Conductin* knockout clones (Cdt KO) (**k**). To facilitate comparisons of the Gαi2 effects in cells with (black bars) and without conductin (siCdt/Cdt KO, red bars) in (**h, i, k**), the initial luciferase activities without Gαi2 were set to 100% for both conditions, and the luciferase activities with Gαi2 are presented relative to the respective initial activity. **j, l** Western blotting showing conductin knockdown and Gαi2 expression in the DLD1 cells used in (**i, j**), and loss of conductin expression in the SW480 knockout clones used in (**k, l**). α-Tubulin: loading control. Results are mean ± SEM (n = 80 [**b**], n = 3 [**c**], n = 5 [**d, f**], n = 4 [**g, h, i, k**]). *p < 0.05, **p < 0.01, ***p < 0.001 (two-sided Student's t-test). Molecular weight is indicated in kDa (**c, e, j, l**). Source data and exact p-values are provided as source data file.

of triggering conductin polymerization by inhibiting aggregation, GBZ treatment reduced the sizes of aggregates formed by endogenous conductin, as detected by native gel electrophoresis (Fig. 6c). Importantly, GBZ treatment decreased β-catenin levels in SW480 cells, which was markedly attenuated by conductin knockout, demonstrating dependency of GBZ activity on conductin (Fig. 6d, e). Moreover, GBZ strongly reduced the expression of β-catenin target genes at the protein level (*AXIN2*, Fig. 6d, f) and mRNA level in colorectal cancer cells (*AXIN2*, *LGR5*, Fig. 6g). Consistently, GBZ inhibited Wnt/β-catenin signaling in

four different cell lines in a dosage-dependent manner, as assessed in luciferase reporter assays (Fig. 6h–j and Supplementary Fig. 13a). Similarly, treatment with other α2-adrenoceptor agonists (clonidine and UK 14,304) inhibited Wnt signaling (Supplementary Fig. 13b, c), indicating a general role of α2-adrenoceptors in controlling the Wnt pathway. As a control, GBZ or clonidine did not reduce the activity of a constitutively expressed luciferase (Supplementary Fig. 13d, e). Of note, GBZ-mediated inhibition of Wnt signaling in colorectal cancer cells (DLD1, SW480) was significantly impaired after conductin

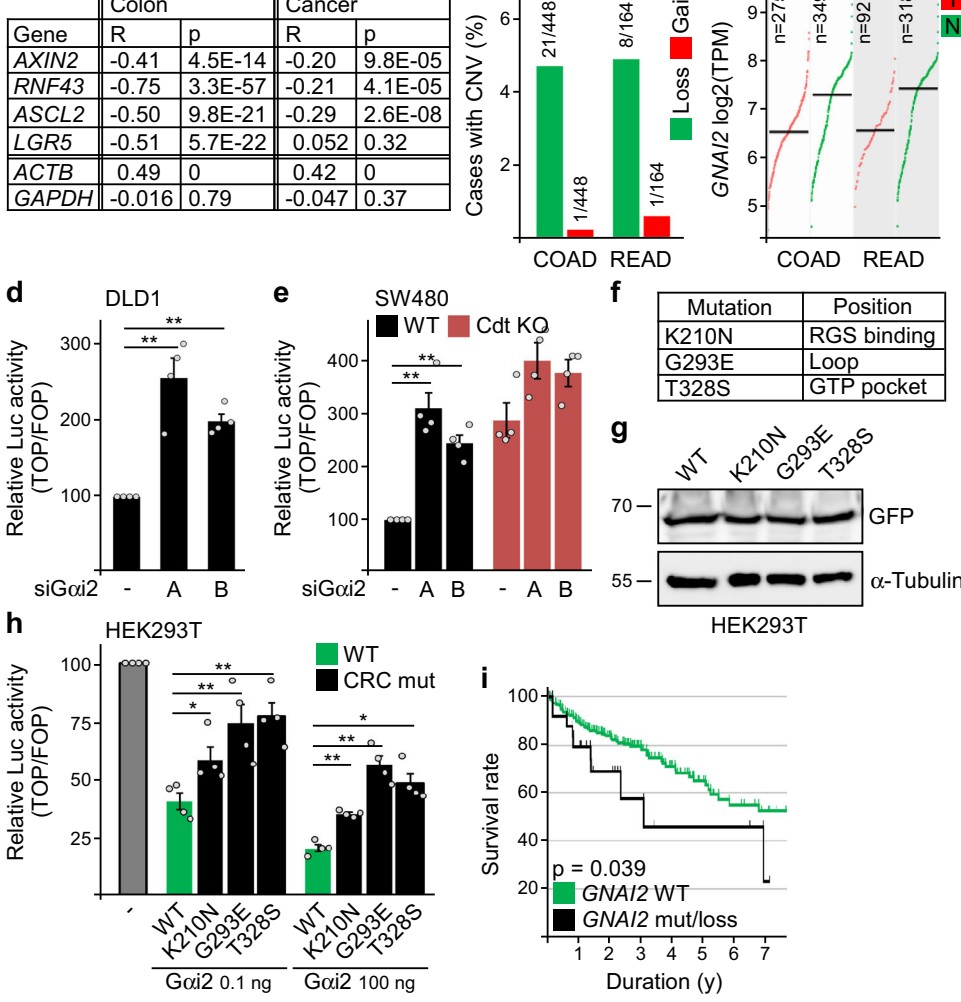

**Fig. 5 Inhibitory Gαi2 aberrations are associated with reduced survival of colon cancer patients. a** Strong negative correlation of *GNAI2* mRNA expression with β-catenin target genes (*AXIN2, RNF43, ASCL2, LGR5*) as shown by the negative Pearson's correlation coefficients (*R*) and *p*-values, and no or positive correlation with housekeeping genes (*ACTB, GAPDH*) in normal human colon tissue and colorectal cancer analyzed with GEPIA[67]. **b** Copy number variation (CNV) of the *GNAI2* gene observed in the TCGA colorectal cancer data sets COAD and READ. **c** *GNAI2* mRNA expression in tumors (T) of the COAD and READ data set compared to normal colon tissue (N). TPM: transcripts per million. **d**, **e** Luciferase activity (TOP/FOP) in DLD1 cells (**d**) and SW480 WT and *AXIN2/Conductin* knockout (Cdt KO) cells (**e**) transfected with two different siRNAs against Gαi2 (A, B). **f** Three *GNAI2* colorectal cancer mutations that were predicted as "deleterious" by Sorting Intolerant From Tolerant (SIFT)[66]. **g** Western blotting for GFP and α-tubulin (loading control) in lysates of HEK293T cells used in (**h**). Molecular weight is indicated in kDa. **h** Luciferase activity (TOP/FOP) in Wnt3a-treated HEK293T cells transfected with rising amounts of WT Gαi2-GFP or colorectal cancer mutants (CRC mut). **i** Survival analysis of colon cancer patients with *GNAI2* missense mutations or copy number loss (n = 25) compared to patients without such *GNAI2* alterations (n = 338) in the COAD data set. Results are mean ± SEM (*n* = 4 [**d**, **e**, **h**]). **p* < 0.05, ***p* < 0.01 (two-sided Student's *t*-test). Source data and exact *p*-values are provided as source data file.

knockdown or knockout (Fig. 6i, j), showing dependency on conductin. Moreover, Gαi2 (Supplementary Fig. 13f, g) or α2a-adrenoceptor knockdown (Supplementary Fig. 13h, i) markedly attenuated Wnt signaling inhibition by GBZ, suggesting specific GBZ function via the known α2-adrenoceptor/Gαi-dependent mechanism rather than off-target effects. Supportively, the specific GBZ function of inhibiting PKA was dosage-dependent within the range of concentrations used for Wnt signaling inhibition (Supplementary Fig. 13j). Together, our data suggest that the GBZ-induced activation of Gαi2 via α2-adrenoceptors triggers conductin condensation, thereby promoting β-catenin degradation and inhibiting Wnt signaling.

**GBZ treatment inhibits colorectal cancer growth**. Functionally, GBZ treatment inhibited the growth of intestinal organoids (Fig. 7a, b), which depends on Wnt signaling[35]. Moreover, GBZ

strongly decreased colony formation of colorectal cancer cells (SW480 and DLD1), which exhibit high endogenous Wnt signaling, but not of U2OS cells with lower endogenous pathway activity (Fig. 7c, d), consistent with specific growth inhibition via Wnt pathway suppression. Consistently, GBZ reduced the numbers of SW480 and DLD1 cells in MTT assays (Fig. 7e, f and Supplementary Fig. 14a, b). Here, knockout or knockdown of conductin markedly attenuated growth reduction demonstrating conductin dependency (Fig. 7e, f and Supplementary Fig. 14a, b). In line with inhibiting a pro-proliferative signaling pathway, GBZ treatment strongly reduced cell divisions of SW480 and DLD1 cells in a dosage-dependent manner, but had almost no effect in SW480 conductin knockout cells (Fig. 7g, Supplementary Figs. 14c and 15). Similar to conductin knockout and knockdown, Gαi2 knockdown significantly rescued GBZ-induced growth inhibition of cancer cells (Fig. 7h, i and Supplementary Fig. 14d, e)

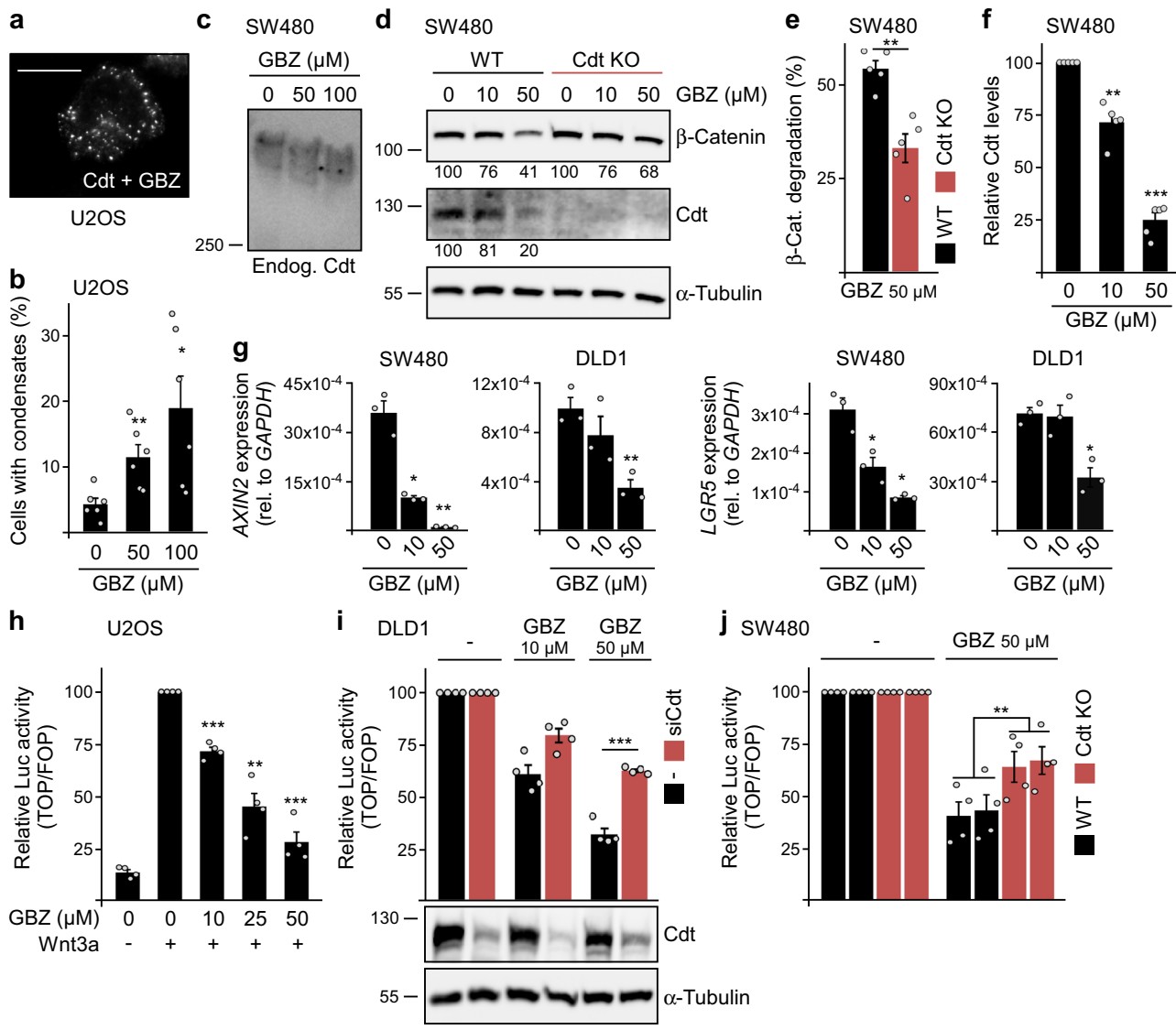

**Fig. 6 Activation of Gαi2 by GBZ treatment inhibits Wnt signaling in colorectal cancer cells. a** Immunofluorescence staining of Flag-Conductin (Cdt) in U2OS cells, which were treated with 50 µM GBZ overnight. Scale bar: 20 µm. **b** Percentage of 1800 transfected cells from six independent experiments as in (**a**) showing conductin condensation. **c** Western blotting for endogenous (endog.) conductin under native conditions in lysates of SW480 cells treated with indicated GBZ concentrations overnight. **d** Western blotting for β-catenin, conductin, and α-tubulin (loading control) in hypotonic lysates of WT SW480 and *AXIN2/Conductin* knockout cells (Cdt KO), which were treated with indicated concentrations of GBZ for 48 h. Numbers below the blots show the relative protein amounts normalized to α-tubulin. **e, f** Quantification of β-catenin degradation (**e**) and expression of the β-catenin target gene conductin (**f**) in WT and knockout cells after GBZ treatment based on five independent experiments as in (**d**). **g** mRNA expression of the β-catenin target genes *AXIN2* and *LGR5* relative to *GAPDH* in SW480 and DLD1 cells treated with indicated concentrations of GBZ. **h–j** Luciferase activity (TOP/FOP) after treatment with indicated GBZ concentrations in U2OS cells without and with Wnt3a stimulation (**h**), in DLD1 cells without and with conductin knockdown (**i**), and in SW480 WT and *AXIN2/Conductin* knockout cells (**j**). Western blots in **i** show the efficiency of conductin knockdown. To facilitate comparisons of the GBZ effects in cells with (black bars) and without conductin (siCdt/Cdt KO, red bars) in (**i, j**), the initial luciferase activities without GBZ were set to 100% for both conditions, and the luciferase activities with GBZ treatment are presented relative to the respective initial activity. Results are mean ± SEM (*n* = 6 [**b**], *n* = 5 [**e, f**], *n* = 3 [**g**], *n* = 4 [**h–j**]). *\**p* < 0.05, *\*\**p* < 0.01, *\*\*\**p* < 0.001 (two-sided Student's *t*-test). Molecular weight is indicated in kDa (**c, d, i**). Source data and exact *p*-values are provided as source data file.

and, consistently, transient Gαi2 expression phenocopied growth inhibition (Fig. 7j and Supplementary Fig. 14f). Together, our data show that GBZ inhibits the growth of colorectal cancer cells in vitro, depending at least partially on the identified Gαi2-conductin-axis. We next tested whether GBZ would inhibit tumor growth in vivo. For this, nude mice developing tumors after subcutaneous injection of SW480 colorectal cancer cells were treated with GBZ via the drinking water, starting when tumors were already palpable. Importantly, GBZ treatment reduced tumor growth by about 60% (Fig. 8a–d). In addition, we used the Min

(multiple intestinal neoplasia) mouse model, in which intestinal tumors develop spontaneously due to mutational inactivation of the negative Wnt pathway regulator APC[36]. Oral GBZ treatment significantly reduced the average tumor size in these mice by about 45% (Fig. 8e), signified by a shift of the frequency distribution toward smaller tumors (Fig. 8f, h) and a marked reduction of highly progressive, large tumors in treated animals (Fig. 8g, h). The total number of tumors was not significantly altered (Fig. 8i), indicating that GBZ did not decrease tumor initiation in our experimental setup, which is in line with the late treatment start at

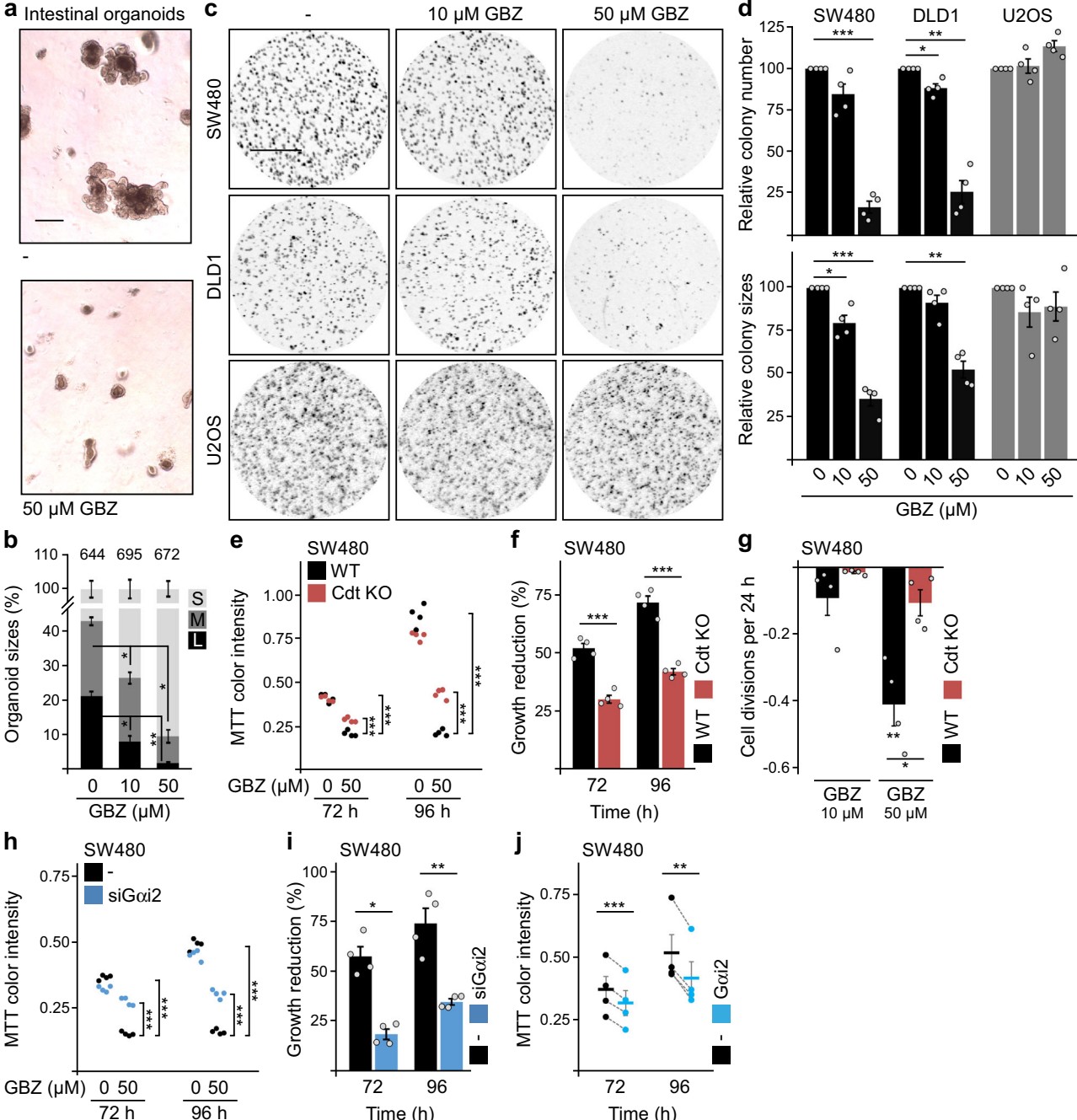

**Fig. 7 GBZ treatment inhibits the growth of colorectal cancer cells. a** Intestinal organoids grown within 120 h in the absence (−) or presence of 50 μM GBZ. Scale bar: 200 μm. **b** Quantification of small (S), medium (M), and large (L) organoids grown in the presence of 0, 10, or 50 μM GBZ in three experiments as in (**a**). Numbers above the bars indicate the total number of intact organoids. **c** Colonies grown from SW480, DLD1, and U2OS cells within 96 h in the presence of 0, 10, and 50 μM GBZ, stained by ethidium bromide and visualized with UV light. Scale bar: 0.5 cm. **d** Automated quantification of colony numbers and sizes from four independent experiments as in (**c**). **e**, **h** MTT color intensity reflects the number of viable SW480 cells growing for 72 and 96 h in the presence of 0 and 50 μM GBZ. WT cells are compared to *AXIN2/Conductin* knockout (Cdt KO) cells (**e**), and control siRNA transfected cells (−) to siGαi2 transfected cells (**h**). Each dot represents a technical replicate (*n* = 4). **f**, **i** Growth reduction by GBZ of four independent experiments as in (**e**, **f**) or in (**h**, **i**) (growth reduction is calculated as the difference between the MTT color intensities without and with GBZ treatment, and presented as a percentage relative to the MTT intensities without treatment). **g** Reduction of cell divisions of SW480 WT and *AXIN2/Conductin* knockout (Cdt KO) cells treated with indicated GBZ concentrations compared to untreated cells, calculated based on four independent CFSE dilution experiments as shown in Supplementary Fig. 15. **j** MTT color intensity reflects the number of viable SW480 cells expressing similar levels of GFP (−, control) or Gαi2-GFP after cell sorting and growth for 72 and 96 h (see Methods for details). The dots represent independent experiments (*n* = 4), and the lines connect values of GFP and Gαi2-GFP expressing cells of the same experiment. Results are mean ± SEM (*n* = 3 [**b**], *n* = 4 [**d**, **f**, **g**, **i**, **j**]). *p < 0.05, **p < 0.01, ***p < 0.001 (two-sided Student's *t*-test). Source data and exact *p*-values are provided as source data file.

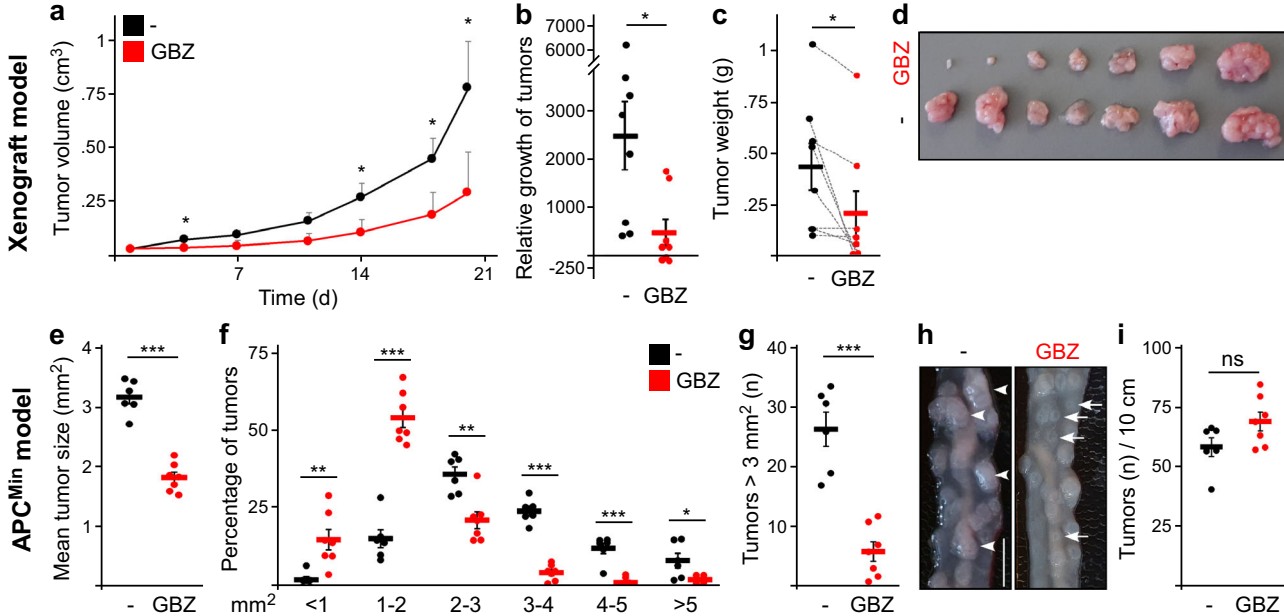

**Fig. 8 GBZ treatment inhibits intestinal tumor growth in vivo. a** Volumes of SW480 xenograft tumors growing in BALB/c nude mice that were untreated ($n = 8$) or treated with 50 µM GBZ in the drinking water ($n = 8$). **b** Growth of the individual tumors (dots) calculated as percentage difference of endpoint tumor volume relative to tumor volume before treatment. Note the shrinkage of three GBZ treated tumors ($n$[untreated] $= 8$, $n$[treated] $= 8$). **c** Endpoint weight measurement of individual tumors (dots). Dotted lines connect the tumors that were paired according to similar sizes and appearance before the treatment ($n$[untreated] $= 8$, $n$[treated] $= 8$, see Methods for details). **d** Excised tumors were ordered to match the pairs of treated (upper row) and untreated tumors (lower row). **e–i** Analysis of intestinal tumors in untreated ($n = 6$, black dots) and GBZ treated ($n = 7$, red dots) APC^Min mice: mean tumor size (**e**), the relative frequency distribution of tumor sizes (**f**), the absolute number of tumors exceeding 3 mm$^2$ per 10 cm intestine (**g**), representative images showing large tumors in untreated animals (arrowheads) and small tumors in treated animals (arrows, scale bar: 0.5 cm) (**h**), and the total number of tumors per 10 cm intestine (**i**). Results are mean ± SEM. *$p < 0.05$, **$p < 0.01$, ***$p < 0.001$ (two-sided Student's $t$-test). Source data and exact $p$-values are provided as source data file.

about six weeks of age, when the mice most likely already harbored tumorous lesions[37]. Consistent with reduced tumorigenesis, GBZ treatment partially rescued splenomegaly (Supplementary Fig. 16a, b), a symptom of APC^Min mice positively correlating with the overall tumor burden[38–40]. Together, our data show that GBZ inhibits Wnt-driven intestinal tumor growth and identify GBZ as a candidate for targeted cancer therapy.

## Discussion

In this study, we characterize Gαi2 as a potent negative regulator of the Wnt/β-catenin signaling pathway. This regulation appeared to be rather universal, as it was observed in various cell lines derived from diverse tissues including colorectal cancer. Mechanistically, our data suggest that binding of Gαi2 to the conductin RGS domain prevents RGS–RGS aggregation via steric interference, thereby inducing conductin polymerization into condensates. Our experiments suggest that enhanced polymerization, in turn, promoted conductin-mediated degradation of β-catenin (Fig. 9), since Gαi2-promoted β-catenin degradation was completely abolished by two distinct mutations of conductin that prevented polymerization based on independent molecular mechanisms (QVL and M3), thereby minimizing the risk of confounding off-target effects. Induction of conductin activity via condensation is consistent with previous studies linking polymerization and activity of axin proteins[7,8,10,31,41]. In line with the identified mechanism, knockout/knockdown experiments revealed a strong dependency of Gαi2-mediated Wnt signaling inhibition on conductin. Thus, feedback regulation of Wnt signaling by conductin will be armed upon Gαi2 activation and more permissive without. In contrast to previously

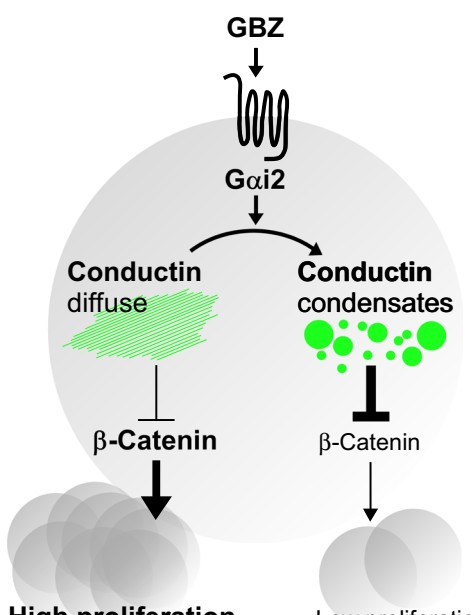

**Fig. 9 Working model for GBZ inhibiting tumor growth.** Activation of Gαi2 by GBZ via G-protein coupled receptors induces polymerization of diffusely distributed conductin into condensates, thereby promoting conductin-mediated β-catenin degradation and, consequently, decreasing cell proliferation.

described axin inhibition by Gαs and Gαo[16–18], our data show activation of the axin homolog conductin by Gαi2, suggesting that GPCR signaling can inhibit or promote β-catenin destruction complex assembly depending on the G-protein. In addition, G-proteins modulate Wnt signaling via the pathway-activating kinase PKA[22,23]. While Gαs supports axin-dependent Wnt pathway activation by activating PKA, we suggest that Gαi2 supports conductin-dependent pathway inhibition by inhibiting PKA. This similarly structured, double-impact mechanism targeting the Wnt pathway at two levels is likely meant to ensure robust pathway regulation indicative of a high physiological relevance. Indeed, Gαi2 activation by GBZ inhibited the growth of intestinal organoids, indicating impaired stem cell proliferation. Moreover, in the intestinal epithelium, Gαi2-coupled α2-adrenoceptors are specifically expressed in stem cells and stimulation with norepinephrine decreased pro-proliferative gene expression[42]. We, therefore, suggest a direct impact of the sympathetic autonomic nervous system on intestinal epithelial renewal via the revealed Gαi2-Wnt signaling axis.

By discovering inducible polymerization of conductin, we add this scaffold protein to the list of polymerizing Wnt pathway regulators, together with axin and dishevelled[43,44]. The striking transition of transiently expressed conductin from a completely diffuse distribution to concentrated condensates upon Gαi2 co-expression or GBZ treatment identified conductin polymerization as a tightly regulated process, which most likely reflects assembly of endogenous β-catenin degradation complexes, as suggested by our functional data. Since any RGS domain-interactor masking the conductin aggregon is predicted to promote polymerization, we are positive that future studies will extend the importance of the aggregon by identifying more aggregon-targeting proteins from different signaling pathways. Notably, we observed key properties of phase-separated biomolecular condensates for conductin puncta, such as temperature sensitivity, concentration dependency, and fusion[11,12]. Thus, our study impacts signaling research beyond the Wnt pathway by describing regulated induction of biomolecular condensates. With the aggregating conductin RGS domain, we reveal an example for an interaction site that counteracts phase separation, adding a new aspect to the phase separation model[12]. Consequently, induction of condensation is not only possible by activating interaction sites, as described[14], but also by inactivating interfering sites, such as the RGS domain in conductin, extending the repertoire of molecular mechanisms controlling condensation.

Importantly, our findings have implications for colorectal cancer development and treatment. Human cancer data indicate silencing of Gαi2-mediated Wnt pathway suppression during colorectal carcinogenesis via loss of GNAI2 copies and inactivating missense mutations. Notably, such GNAI2 alterations are associated with reduced patient survival. Thus, our data point to a tumor suppressor function of Gαi2 via inhibition of Wnt signaling. Consistently, Gαi2-deficient mice develop adenocarcinoma in the colon[45]. Although these cancers seem to develop as a consequence of preceding colitis, Rudolph et al. already hypothesized direct tumor suppression by Gαi2, since there are other gene-deficient mice developing colitis without tumors. As Wnt/β-catenin signaling is activated in response to colitis for tissue regeneration[46], we believe that Gαi2 restricts pathway activation to sub-oncogenic levels by enhancing the conductin-driven negative feedback, while losing of Gαi2 results in reduced conductin activity, hyperactive Wnt signaling, and cancer growth. Together, our identified mechanism, patient data, and the phenotype of Gnai2 knockout mice suggest a tumor suppressor function of Gαi2 in colorectal cancer. Inactivating GNAI2 alterations occur in about 6% of colorectal cancer patients, which is about the frequency of oncogenic β-catenin (5%) or conductin

(7%) mutations but far below the loss of APC (77%), the major tumor suppressor of the Wnt pathway[2]. As inactivating GNAI2 mutations are associated with a severe decrease of the three-year survival rate, these genetic alterations might become of value as a prognostic marker. In the other 94% of the patients, activation of Gαi2 by the specific α2-adrenoceptor agonist GBZ holds potential for therapy. GBZ inhibited Wnt signaling in colorectal cancer cells, and knockdown/knockout experiments showed the dependency of GBZ-mediated Wnt signaling inhibition on Gαi2 and conductin, suggesting that GBZ activates Gαi2 to promote conductin-mediated β-catenin degradation. Attenuated inhibition of Wnt signaling after α2a-adrenoceptor knockdown suggests a specific function of GBZ as α2-adrenoceptor agonist rather than off-target effects. Supportively, other α2-adrenoceptor agonists (clonidine, UK 14,304) similarly inhibited the Wnt pathway. In line with inhibiting a pro-proliferative pathway, GBZ treatment reduced the proliferation of colorectal cancer cells in a conductin-dependent manner in vitro (Fig. 9).

Of note, oral administration of GBZ via the drinking water reduced tumor growth in a xenograft mouse model for colorectal cancer and in APC$^{Min}$ mice, which spontaneously develop intestinal tumors. As the xenograft transplanted mice and most likely also the APC$^{Min}$ mice already carried tumors when we started to administer GBZ, our experimental setup comes close to the treatment of cancer patients. Tumor growth was inhibited by 50 μM GBZ in the drinking water, which the mice consumed ad libitum. Based on a daily water intake of 6 ml[47], the mice took up 0.3 μmol GBZ, corresponding to a daily dose of 3.47 mg/kg for a 20 g mouse. Notably, comparable doses were used in rats in the first study on GBZ as a hypotensive agent[48]. The human equivalent dose of 3.47 mg/kg in mice calculates to 0.28 mg/kg, applying the FDA recommended dose conversion, which is based on body surface area[49]. Thus, a colorectal cancer patient of 60 kg (FDA standard human weight) requires 16.8 mg GBZ per day, which is within the daily GBZ doses used to treat hypertension ranging from 8 to 64 mg[50]. We, therefore, believe that the FDA-approved drug GBZ, which has been safely used over decades to treat hypertension, holds potential for targeted colorectal cancer therapy.

## Methods

**Cell culture, transfection, and treatment**. DLD1, HEK293, HEK293T, U2OS, and SW480 cells were cultured in low glucose Dulbecco's Modified Eagle's Medium (DMEM) supplemented with 10% fetal calf serum (FCS) and antibiotics at 37 °C in a 10% CO$_2$ atmosphere, and passaged according to ATCC recommendations. Transfections were performed using polyethylenimine (for plasmids in HEK293, HEK293T, and U2OS), Lipofectamine2000 (Invitrogen, for plasmids in DLD1 and SW480, and for plasmids together with siRNAs), or Oligofectamine (Invitrogen, for siRNAs) according to manufacturer's instructions. Wnt3a medium was prepared as originally described[51]. For AlF$_4^-$ treatment, cells were incubated in Tyrode's solution without (control) or supplemented with AlCl$_3$, NaF, and MgCl$_2$[52]. Bt$_2$cAMP (D0627), clonidine (C7897), guanabenz (GBZ, G110)[48], and UK 14,304 (U104) were obtained from Sigma-Aldrich, and forskolin (ab120058) and H-89 (ab143787) from Abcam.

**Genetically engineered cell lines**. To generate U2OS cells stably expressing GFP-Conductin, cells were transiently transfected, re-seeded at low density, and cultured in the presence of 500 μg/ml G418 for more than one week. Cell colonies, which clonally grew due to genomic integration of the expression vector including a neomycin resistance cassette, were isolated and assessed for expression of GFP-Conductin by immunofluorescence and Western Blot. HEK293 cells stably expressing shRNAs against Gα proteins were obtained similarly using selection with puromycin (10 μg/ml), and the shRNA sequences are provided below. CRISPR/Cas9-generated SW480 AXIN2/Conductin knockout cells have been described previously[7].

**Intestinal organoid growth assay**. Intestinal organoids were derived from crypt fragments, which were mechanically released from the small intestine of a six-week-old C57BL/6 mouse and enriched by size exclusion of villi fragments[35]. Organoids grew embedded in matrigel (Corning) in Advanced DMEM/F-12

medium supplemented with GlutaMAX, B-27, N-2 (ThermoFisher), 10 mM HEPES, 1 mM N-acetylcysteine (Sigma-Aldrich), 0.1 µg/ml mEGF (Immuno-Tools), 0.1 µg/ml mNoggin (PeproTech), 20% R-spondin1-conditioned medium and antibiotics at 37 °C in a 10% $CO_2$ atmosphere, and were passaged weekly. R-spondin1 medium was prepared from 293 T cells stably expressing HA-mR-Spondin1-Fc[53]. For the organoid growth assay, the medium of freshly passaged organoids was supplemented with 0, 10, or 50 µM GBZ, and after 120 h small (0–1 buds), medium (2–4 buds), and large (>4 buds) organoids were counted. The assay was performed in technical octuplicates.

**Cell growth assays.** For colony formation assays, 2500 cells were seeded per well of a six-well plate and grown in medium with 1% FCS in the presence of 0, 10, and 50 µM GBZ for 96 h. Then, cells were fixed in 3% PFA, stained with ethidium bromide (50 µg/ml in PBS), and visualized in a UV gel documentation system (Herolab), where images were acquired at constant settings (E.A.S.Y. Win32 software, Herolab). From these images, colony numbers and sizes were quantified in an automated fashion as light objects using the MetaMorph analysis software (Molecular Devices). For MTT assays, 2000 cells per well of a 96-well plate were seeded or re-seeded 24 h after siRNA transfection, when required, and grown in medium with 1% FCS and 0 or 50 µM GBZ for 72 or 96 h. In case of transient expression experiments, cells with similar GFP (control) or Gαi2-GFP expression were sorted based on the GFP fluorescence and seeded as 2500 cells per well of a 96-well plate (MoFlo XDP cell sorter, Beckman Coulter), and grown in medium with 1% FCS for 72 or 96 h. Then, cells were incubated with 0.5 mg/ml MTT (Sigma-Aldrich) for 4 h at 37 °C to allow MTT cleavage in vital cells, and the cleaved MTT formazan was dissolved and measured in a Spectra MAX 190 with SoftMax Pro software (Molecular Devices), according to manufacturer's instructions. The measured formazan absorbance is directly proportional to the number of living cells. This assay was performed in technical quadruplicates (knockout and knockdown experiments), or quintuplicates (transient expression experiments).

**Cell proliferation assay.** Cell proliferation was investigated in carboxyfluorescein succinimidyl ester (CFSE) dilution experiments. Cells were pulse-labeled with CFSE and grown in the presence of 0, 10, and 50 µM GBZ. After 72 h, the remaining CFSE fluorescence (green) was measured by FACS (FACSCalibur with CellQuest software, BD Bioscience) in 50,000 cells per sample. Every cell division reduces the CFSE fluorescence intensity by 50% since CFSE is equally distributed between both daughter cells. For instance, a twofold higher CFSE fluorescence intensity means one cell division less in 72 h, which allows calculating how strong GBZ treatment affected cell proliferation (Fig. 7g and Supplementary Fig. 14c).

**Expression vectors and interfering RNAs.** Expression vectors encoding for R-FlincA[33], RGS6xHis-Gαi2[54], 6xHis-RGS10[55], Gαo-GFP, Gαo-GST-HA[56], Flag-β-Catenin, Flag-Cdt, Flag-CdtM3[27], HA-Cdt, HA-Cdt 2-345, GFP, GFP-Cdt, GFP-Cdt QV (Q188P V189S), GFP-Cdt QVL (L99R Q188P V189S), GFP-Cdt 2-345, GFP-Cdt 2-345 QV (Q188P V189S) are published[7]. Flag-Cdt QVL (L99R Q188P V189S), GFP-RGS^Cdt (Conductin aa 71-211; O88566), GFP-RGS^Cdt QVL (L99R Q188P V189S), Gαi1-GFP, Gαi2-GFP, Gαi3-GFP, Gαi1-GST-HA, Gαi2-GST-HA, Gαi3-GST-HA, Gαi2-HA, Gαq-GFP, Gαs-GFP, Gα12-GFP, GST-RGS^Cdt (Conductin aa 67₋210), GST-Cdt 2-210, RGS6xHis-RGS^Cdt (Conductin aa 67–210) and T7 expression vectors for in vitro protein synthesis (Gαi2-GFP, Gαi2-GFP K210N, Gαi2-GFP G293E, Gαi2-GFP T328S) were cloned by standard molecular biology methods, and point mutations were introduced via site-directed mutagenesis: GFP-Cdt 2–345 QVL (L99R Q188P V189S), GFP-Cdt 2-345L (L99R), Gαi2-GFP G184S, Gαi2-GFP K210N, Gαi2-GFP G293E, Gαi2-GFP T328S. All generated expression vectors were verified by sequencing. The siRNAs targeting human conductin/axin2 (5′-GAGAUGGCAUCAAGAAGCA-3′) and Gαi2 (A: 5′-CCAGCUACAUCCA-GAGUAAdTdT-3′, B: 5′-GGAGCGUAUUGCACAGAGU-3′) have been used previously[7,57,58]. For α2a-adrenoceptor knockdown, the ON-TARGETplus Human ADRA2A siRNA SMARTpool from Dahrmacon was used. The following shRNA sequences were used for the stable HEK293 cells, shGαi1 (5′-cccGTGGAAAGA TAGTGGTGTAttcaagaga TACACCACTATCTTTCCACttttt-3′, shGαi2 (A: 5′-cc cGCAAAGACACCAAGGAGATtcaagagaATCTCCTTGGTGTCTTTGCtttt-3′, B: 5′-ccgCCAGCTACATCCAGAGTAAttcaagagaTTACTCTGGATGTAGCTGG ttttt-3′), shGαi3 (A: 5′-cccGCTTATGGTTGAAGGTTAAttcaagagaTTAACCTTCA ACCATAAGCttttt-3′, B: 5′-cccGGAGATTTGTGATGGATAA ttcaagagaTTATC CATCACAAATCTCCttttt-3′).

**Immunofluorescence.** Cells, which were transfected and/or treated as indicated in the figures, were fixed in 3% PFA (Supplementary Fig. 2i) or methanol (others), permeabilized with 0.5% Triton X-100, blocked with medium to reduce unspecific antibody binding, and subsequently incubated with primary and fluorochrome-conjugated secondary antibodies (see below). Analysis and image acquisition was performed at an Axioplan II microscope system (Carl Zeiss) using a Plan-NEOFLUAR 100×/1.30 NA oil objective and a SPOT RT Monochrome camera (Diagnostic Instruments). Cells were classified according to the conductin distribution as "diffuse", "with condensates" or "with membrane recruitment" by manual counting. For stringent classification, cells displaying up to five condensate-like structures were classified as "diffuse", since even some of the cells with largely

diffuse conductin distribution showed a low number of condensate-like structures possibly including centrosomes[59]. Cells displaying more than five condensate-like structures were classified as "with condensates". Only cells with a uniform conductin staining covering at least 50% of the plasma membrane were classified as "with membrane recruitment". For experiments with smaller differences, quantification was performed in a blinded fashion. Numbers of condensates per cell and cell areas for subcategorization of cells with condensates were determined via image-based analysis using the Spot Detector tool of the Icy open-source bio-imaging software (Institut Pasteur, version 2.2.1.0)[60]. Fluorescence intensities were quantified from images acquired at constant exposure times by the MetaMorph analysis software (Molecular Devices), when required.

**Live-cell imaging.** Time-laps microscopy of GBZ treated U2OS cells stably expressing GFP-Conductin was performed at 23-24 °C using an Axiovert 200 microscope system (Carl Zeiss) with a Plan-NEOFLUAR 40×/1.30 NA oil objective and a xiD CCD camera (Ximea). Images were acquired at constant exposure times every 3 min with Micro-Manager 1.4 (Vale Lab, UCSF), and movies were rendered using Photoshop 19.1.5 (Adobe).

**Western blot.** Cells were lysed in Triton X-100 containing buffers (150 mM NaCl, 20 mM Tris-HCl pH 7.5, 5 mM EDTA, 1% Triton X-100, Roche protease inhibitor cocktail, or the luciferase assay buffer stated below), in an SDS containing buffer (100 mM NaCl, 5 mM $MgCl_2$, 20 mM Tris-HCl pH 7.5, 2 mM EDTA, 1% Triton X-100, 0.5% SDS, 10% glycerin, protease inhibitor cocktail) for better solubilization of Gα proteins, or in hypotonic lysis buffer (20 mM Tris-HCl pH 7.5, 1 mM EDTA, Roche protease inhibitor cocktail) when assessing β-catenin levels. Protein samples were separated via polyacrylamide gel electrophoresis under denaturing or native conditions (no SDS), when indicated, and transferred onto a nitrocellulose membrane, which was probed with indicated primary and respective horseradish peroxidase (HRP)-conjugated secondary antibodies (see below). Protein bands were detected via light emission upon HRP-catalyzed oxidation of luminol in a LAS-3000 with Image Reader software (FUJIFILM) or a PXi6 (Syngene) with GeneSys software, and band intensities were quantified with AIDA 2D densitometry or Fiji-ImageJ when required.

**Antibodies.** Primary antibodies: rb α Axin2 [WB: 1:1000], 2151S; m α GST [WB: 1:1000], 2624S CellSignaling/rb α GFP [WB: 1:2000], GTX113617 GeneTex/rb α Gαi3 [WB: 1:2000], 371726 Merck/m α RGS-His [WB: 1:2000], 34650 Qiagen/m α GFP [WB: 1:1000], 11814460001; rat α HA [WB: 1:2000], 11867423001 Roche/m α β-catenin [IF: 1:200, WB: 1:1000], sc-7963; m α Gαi1 [WB: 1:50], sc-13533; m α Gαi2 [WB: 1:50–1:200], sc-13534 Santa Cruz Biotechnologies/rat α α-tubulin [WB: 1:1000], MCA77G Serotec/m α β-actin [WB: 1:1000], A5441; rb α Flag [IF: 1:300, WB: 1:1000], F7425; rb α HA [IF: 1:200, WB: 1:1000], H6908; m α α-tubulin [WB: 1:2000], T6199 Sigma-Aldrich.

Secondary antibodies: goat α mouse/rabbit-Cy3 [1:300], goat α mouse/rabbit/rat-HRP [1:1000–1:2000] (Jackson ImmunoResearch).

**Expression and purification of recombinant proteins.** Protein expression was induced with 1 mM IPTG in transformed Rosetta Gami (D3) *Escherichia coli* (Novagen) in their logarithmic growth phase at 17 °C overnight. Bacteria were lysed in TBS buffer (20 mM Tris-HCl pH 7.5, 150 mM NaCl) with 1 mM PMSF in a One-shot Cell disruptor (Constant Systems). Recombinant His-tagged and GST-tagged proteins were precipitated from precleared lysates (15,000*g*/15 min) on Ni-NTA beads (Pierce) and glutathione beads (GE Healthcare), respectively. Precipitated proteins were washed on beads and eluted using 300 mM imidazole (His-tag) or 20 mM reduced glutathione (GST-tag). For His-Gαi2, washed beads were incubated in TBS buffer containing 0.1 mM DTT, 10% glycerol, 100 µM GDP, and 5 mM $MgCl_2$ at 4 °C on a rotary shaker overnight prior to elution. Purified proteins were rebuffered in TBS by at least a 10,000-fold exchange using centrifugal concentration units (Amicon).

For in vitro protein synthesis of Gαi2-GFP and the cancer mutations K210N, G293E, and T328S, the PURExpress In Vitro Synthesis Kit (NEB) was used, according to the manufacturer's guidelines.

**GST pulldown.** Cells co-transfected with GST-tagged Gα proteins and GFP-RGS^Cdt were lysed (1% Triton X-100 and 10% glycerol in PBS, protease inhibitor cocktail), and the GST-tagged G-proteins were precipitated on glutathione beads (GE Healthcare). After washing, the amounts of precipitated GST-Gα proteins and co-precipitated GFP-RGS^Cdt were assessed and quantified by Western blotting.

For pull-down with in vitro synthesized Gαi2-GFP proteins, the Gα proteins were incubated with purified recombinant GST-RGS^Cdt in PBS containing 1% Triton X-100, 10% glycerol, and 10 µM GTP, and the pulldown was performed as described above.

In case of purified recombinant proteins, His-Gαi2 was preloaded in TBS buffer (see protein purification) supplemented with 10 µM GDP alone or together with 30 µM $AlCl_3$ and 5 mM KF, or with 10 µM GTPγS for 1 h at RT. Then, 1 µM of preloaded His-Gαi2 was incubated with 1 µM of indicated GST proteins in TBS buffer with 0.1% bovine serum albumin for 1 h at RT, before adding glutathione beads (GE Healthcare) for a final incubation of 1 h on a shaker. Glutathione-

interacting proteins were precipitated on the beads, washed, and eluted. The amounts of precipitated GST proteins and co-precipitated His-Gαi2 were determined on Ponceau-stained gels and via Western blotting, respectively.

**BODIPY-FL-GTP GAP assay**. For analysis of GAP activity, 10 μM of indicated protein was mixed with 1 μM His-Gαi2 in TBS buffer (see protein purification). After 10 min preincubation, the fluorescent GTP derivative BODIPY-FL-GTP was injected into the mixture to a final concentration of 1 μM together with 5 mM MgCl$_2$. The kinetics of in vitro Gαi2-GTP binding and hydrolysis were measured as changes in BODIPY-FL-GTP fluorescence intensity, which increases upon G-protein binding and decreases due to GTP hydrolysis, using the Infinity M200Pro multiwell reader with i-control software (Tecan). Based on these measurements, the rate constant of GTP hydrolysis (apparent $k_{hydr}$) was calculated[54,61].

**Sucrose density ultracentrifugation assay**. Transfected HEK293T cells were lysed in a Triton X-100-based buffer (see Western blot), and the lysates were loaded on a linear sucrose density gradient, which had formed via equilibration between a 50% (w/v) and a 12.5% sucrose solution in a 13 × 51 mm centrifuge tube (Beckman Coulter) upon horizontal incubation for 3 h at RT[62]. After centrifugation (201,000g/16 h/25 °C) in an Optima MAX Ultracentrifuge using an MLS-50 rotor (Beckman Coulter), 20 fractions of equal volume were collected, and every second fraction was analyzed by Western blotting (see above).

**Luciferase reporter assay**. Cells were transfected with a luciferase expression vector (TOP-Flash [TCF optimal]: β-catenin-dependent expression, FOP-Flash [TCF far-from-optimal]: β-catenin-independent expression, CRE reporter: PKA activity-dependent expression, or pGL3-Basic: constitutive expression) together with β-galactosidase, co-transfected and/or treated as indicated in the figures, and lysed in luciferase assay buffer (25 mM Tris-HCl pH 8, 2 mM EDTA, 5% glycerol, 1% Triton X-100, 20 mM DTT). Luciferase and β-galactosidase activities were assessed via light emission upon luciferin decarboxylation in a Centro LB 960 Microplate Luminometer (Berthold Technologies) with MikroWin 2000 software and as the release of yellow ortho-nitrophenol upon ortho-nitrophenyl-β-galactoside hydrolysis using a Spectra MAX 190 with SoftMax Pro software (Molecular Devices), respectively. Luciferase activities were normalized to β-galactosidase activities to correct for transfection variances. For TOP/FOP assays, TOP values were subsequently normalized to FOP values to correct for β-catenin-independent, unspecific changes (Microsoft Excel). All assays were performed in technical duplicates.

**qRT-PCR analysis**. To quantify mRNA expression of β-catenin target genes, whole cellular mRNA was extracted (RNeasy mini kit, Qiagen), transcribed into cDNA (AffinityScript QPCR cDNA Synthesis Kit, Agilent Technologies), and the relative abundance of transcripts from individual genes was determined by SYBR Green-based qPCR (CFX96 Real-Time System, Bio-Rad) with gene-specific primer pairs (*AXIN2*: 5′-CCTCAGAGCCGATGGATTTCGGG-3′, 5′-CCAGTTCCTCTCAG-CAATCGGC-3′; *GAPDH*: 5′-GTCAAGGCTGAGAA CGGGAAGC-3′, 5′-GGACTCCACGACGTACTCAGGCG-3′; *LGR5*: 5′-CTTCCAACCTCAGCGTCTT-CACC-3′, 5′-GTCAGAGCGTTTCCCGCAAGAC-3′) in technical triplicates. Expression of β-catenin target genes (*AXIN2, LGR5*) is presented relative to the housekeeping gene *GAPDH* to correct for minor variances in cell numbers between the analyzed samples.

**Mouse tumor models**. For the xenograft model, eighteen seven-week-old, female BALB/c-Nude (CAnN.Cg-Foxn1[nu]/Crl) mice (Charles River) were injected with 5 × 10$^6$ SW480 cells subcutaneously at the right flank. The mice were housed in a UNIPROTECT airflow cabinet (ZOONLAB) at 23 °C, 60% humidity, and a light–dark cycle of 12 h each. Ten days post-injection, the mice were distributed in eight matching pairs according to tumor sizes, and one animal per pair was treated with 50 μM GBZ in the drinking water for 20 days. Tumor growth was monitored over time via repeated caliper measurement, and the tumor volume was calculated as length × width × ½ width. In the end, tumors were excised, weight and volumes were calculated as length × width × height.

For the APC$^{Min}$ mouse model (C57BL/6J-Apc$^{Min}$/J, Charles River), about six-week-old female mice heterozygous for the Min allele were assigned to either the untreated control group or the GBZ treatment group, which was treated orally with 50 μM GBZ in the drinking water. The mice were housed at 21 °C, 53% humidity, and a light–dark cycle of 12 h each. At about 15 weeks of age, the mice were sacrificed, the small intestines were excised and cut into six pieces of similar length. Pieces four and five (counting from the oral side) were opened longitudinally, images were acquired together with a size standard, and the sizes of the individual tumors were determined in mm$^2$ from these images using the MetaMorph analysis software (Molecular Devices). Similarly, lengths and widths of the spleens were determined in cm from images, and spleen volumes were calculated as (½ width)$^2$ × π × length.

The study protocol was approved according to the Tierschutzgesetz in Germany by the Regierung von Unterfranken (55.2.2–2532.2–923), and we have complied with all relevant ethical regulations. The permitted maximal tumor size (diameter > 1.5 cm) was not exceeded.

**R-FlincA-based cAMP measurement**. U2OS cells were transfected with R-FlincA alone or together with Gαi2 at 37 °C for 24 h, incubated at 32 °C for the next 24 h, as described for fluorescence cAMP sensors[33,63], and treated with 1 mM Bt$_2$cAMP overnight, if indicated. Then, the red fluorescence intensity was measured by FACS (FACSCalibur with CellQuest software, BD Bioscience) in 50,000 cells per sample.

**Structural models**. The model structure of the conductin RGS domain was calculated by SWISS-MODEL using the crystal structure of the highly similar axin RGS domain (PDB ID: 1DK8) as template[64,65]. Structural alignments of RGS domains and presentation of the structures were performed with PyMOL (version 1.8. Schrödinger, LLC).

**Analysis of TCGA/GTEx data sets**. To analyze the expression and mutation of *GNAI2* (Gαi2) in colorectal cancer we used the well-curated and carefully-collected data of the TCGA datasets COAD and READ[2], which are provided by the National Cancer Institute (NCI). Analyses of copy number variations (CNV), cancer mutations, and survival were performed using the NCI Genomic Data Commons (GDC) Data Portal. For experimental follow-up, we chose all detected Gαi2 missense mutations that were rated "deleterious" by the Sorting Intolerant From Tolerant algorithm[66]. For the survival analysis, colon cancer patients from the COAD data set with *GNAI2* missense mutations (5 cases) or copy number loss (21 cases) were combined (26 cases of which 25 had the required data for survival analysis) and compared to patient cases with available CNV data and simple somatic mutation data that did not show *GNAI2* alterations (357 cases of which 338 had the required data for survival analysis) using the GDC Data Portal analysis tool (NCI). For analyzing the mRNA expression data provided with the COAD, READ, and The Genotype-Tissue Expression (GTEx, Colon-Sigmoid, and Colon-Transverse) datasets, GEPIA was used[67,68].

**Statistic**. To probe the datasets for statistical significance of differences two-tailed Student's *t*-tests were performed in a non-paired (Figs. 4b, k, 6j, 7e, h, 8b, e, f, g, i, and Supplementary Figs. 8b, c, 14a, d and 16a) or paired fashion (all other experiments), depending on the experimental setup (Microsoft Excel). Based on the nature of the assays and graphical assessment, we assumed normal distribution of the data, which, however, was not formally tested due to the small sample sizes. P-values for the correlation analyses were calculated by the test statistic ($t = R$*square root$((n − 2)/(1 − R^2))$), with $n − 2$ degrees of freedom (Supplementary Fig. 4a), or provided as part of the GEPIA analysis in case of mRNA expression correlations[67]. Differences in patient survival (Fig. 5i) were tested for statistical significance using a log-rank test included in the GDC Data Portal analysis (NCI). For all experiments, n values are explicitly stated in the figure legends, and statistical significance is indicated by asterisks in the figures (*$p < 0.05$, **$p < 0.01$, ***$p < 0.001$), when required.

**Reporting summary**. Further information on research design is available in the Nature Research Reporting Summary linked to this article.

## Data availability

The crystal structure data used in this study are available in the PDB database under accession codes 1DK8, 1EMU, 2GTP, 2IHB, 2IK8, and 2ODE. The human cancer data used in this study are available in the NCI Genomic Data Commons Data Portal database under accession codes COAD and READ. The other mRNA expression data used in this study are available in the GTEx Portal database under accession codes colon-sigmoid and colon-transverse.

The source data underlying Figs. 1d–f, 2a–f, h–k, 3d–g, 4b–l, 5d, e, g, h, 6b–j, 7b, d–j, 8a–c, e–g, i and Supplementary Figs. 2c–e, g, h, j, k, 3b, c, 4a–e, g, 5a, c–e, 6c–e, 7a–l, 8b–k, 9a–e, 11a–g, 12a, b, f, 13a–j, 14a–f and 16a are provided as a Source Data file. All the other data supporting the findings of this study are available within the article and its Supplementary Information. A reporting summary for this article is available as a Supplementary Information file. Source data are provided with this paper.

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

## Acknowledgements

The authors thank Kazuki Horikawa for providing the R-FlincA expression vector, Christoph Becker and Calvin Kuo for providing the HA-mR-Spondin1-Fc cells, the core-unit Cell-sorting with Immunomonitoring (Friedrich-Alexander University Erlangen-Nürnberg), Laura Gardill for initial experimental contributions, and Gabriele Daum and Sabina Troccaz for excellent technical assistance. This study was funded by a grant from the Wilhelm Sander-Stiftung to D.B.B. (2018.017.1), and grants from the Friedrich-Alexander University Erlangen-Nürnberg Interdisciplinary Center for Clinical Research to D.B.B. (J58) and to J.B. and D.B.B (D30) and from the Deutsche Forschungsgemeinschaft to J.B. (BE 1550/12-1).

## Author contributions

C.M., G.P.S., A.K., M.B., and D.B.B. performed the experiments. C.M., G.P.S., A.K., V.L.K., and D.B.B. designed the experiments and interpreted the data. J.B. and D.B.B. conceived the study and wrote the paper, with input from the co-authors.

## Funding

## Competing interests

The authors declare no competing interests.
