## [Peer Review File · Nature Communications]

G α i2-induced conductin/axin2 condensates inhibit Wnt/ β -catenin signaling and suppress cancer growthREVIEWER COMMENTS

Reviewer #1 (Remarks to the Author):

Miete and colleagues describe a novel mechanism that regulates the function of Axin2 (Conductin) as a suppressor of b-catenin signaling through heterotrimeric G proteins. The premise is that Galphai2 (and to a lesser extent of the Galphai proteins) bind to the RGS domain of Axin2 to allow its aggregation in cytoplasmic puncta and also to favor its function as a suppressor of b-catenin signaling. One of the strengths of the work is the identification of an FDA-approved agonist (GBZ) of GPCRs that activate G proteins as a potential anti-colorectal cancer treatment that presumably operates through the Gai2-Axin2 mechanism proposed here. This last part of the manuscript on the action of GBZ is convincing and potentially impactful. The area that, in my opinion, would need significant improvement is on the molecular mechanism of Gai-mediated regulation of Axin2. The following points are listed in approximate order of importance.

- 1- The authors need to provide more definitive data on the requirement for Gai2-Axin2 binding for the regulation of Axin2 localization and function. Evidence so far is correlative and to establish definitely the mechanism the authors will need a mutant that specifically disrupt the Gai2-Axin2 interaction. For clarity, the use of AIF4- gives indirect evidence and might have many indirect consequences, and the use of Axin2 mutants with swapped aggregation positions does not directly address the need for Gai2 binding.
- 2- The interaction between Gai2 and Axin2 needs to be characterized more thoroughly, including experiments with purified components rather than using cellular expression. One specific point is the characterization of Axin2-RGS binding as function of the activation status of Gai2. The authors imply that the interaction occurs with the active Gai2, but this is at odds with results presented in Figure 2A, B, in which cellular Gai2 will be predominantly inactive. Loading with GDP, GTPγS and AIF4- should be compared.
- 3- If Axin2 binds to active Gai2 or to the AIF4-bound mimic of the active transition state, it might be acting as a GAP. This should be evaluated.
- 4- The authors should expand the G protein subtypes investigated. They have only investigated members of the Gi/o family, but Gs, G12 or Gq family Galpha subunits may also bind to Axin2.
- 5- A significant limitation of the mechanistic studies is the overreliance on overexpression systems. There are no experiments to document that endogenous Axin2 behaves like the overexpressed one. Key experiments should be performed to demonstrate that endogenous Axin2 shifts distribution upon manipulation of endogenous Gai2.
- 6- Claims implying that Axin2 distributes to phase separated condensates and that phase separation is important for function should be omitted. The evidence demonstrating that Axin2 is in phase separated condensates is weak for the overexpressed protein and absent for endogenous one, and the evidence to support a causal relationship between phase separation and function is non-existent. There are only correlations between subcellular localization and function.
- 7- Figure 3F, a control showing that Wnt activity can be further increased is missing.
- 8- What is the impact of the Gai2 mutants described in Figure 4 on Axin2 binding and in regulating Axin2 localization?
- 9- I suggest removing "severe" when referring to the differences in patient survival described in the manuscript. The differences are barely statistically significant. This is in part due to the low frequency of genomic Gai2 alterations. The low prevalence of these alterations should also be explicitly discussed (ie, this might not be a driver event like other alterations of the pathway are).

Reviewer #2 (Remarks to the Author):

In the work by Miete et al. the authors show that Gai2 induces condensation of AXIN2/conductin, which is mediated by targeting an aggregon in the RGS domains and leads to the reduction of Wnt signaling. Furthermore, the authors demonstrate the levels of Gai2 are frequently reduced in CRC. Using the Gai2 activating drug guanabenz, the authors use several models of colorectal cancer to show that this drug can diminish Wnt levels and reduce growth of CRC. While the molecular aspect of Gai2/conductin interaction is convincing, the cancer-related focus of the manuscript needs revision. Specifically, it is unclear to which extent guanabenz works through Gai2 to reduce growth of CRC cell lines.

Overall, this is an interesting manuscript building on the e 2019 Nat Comm publication by Bernkopf et al. and providing mechanistic insights into aggregon regulation of axin2/conductin through Gai2 and condensate regulation in general.

Major comments:

As the authors suggest that antiproliferative effects of GBZ functions (partly) through Gai2, the question is if overexpression of Gai2 can phenocopy the effect of GBZ and if siGai2 can rescue tumor cells from the antiproliferative effects of GBZ? The authors should provide additional evidence to show that GBZ works through Gai2.

Minor comments:

Figure 1F: could be transferred to the supplement

Figure 2B: In this figure, it is unclear why Gai2 is only compared to Gai0 and not to the other Gai

Figure 2L: since the approach is not easy to understand, the model in L should be presented before H-K

"Wnt/ β -catenin signaling by promoting conductin condensation in a broad variety of different tissues including colorectal cancer" > this sentence should be rephrased as the authors showed the phenotype in 3-4 different cell lines from 3 tissue backgrounds, and "a broad variety of different tissues" seems inappropriate.

Figure 3H and I: The normalization of -/siCdt seems odd in these sub-figures. As shown in 3F, siCtrl activates TOP Flash reporter activity in a strong fashion, but the way the data is presented in H and I, it seems that there is no difference between the two conditions

Figure 3K: it is not clear in this figure if knockout of conductin increases basal Wnt levels. The current presentation of the data suggests that there is no difference. Also labelling as -/- is confusing, the use of sgCdt or Cdt KO would be easier to comprehend.

"Gai2 functions as tumor suppressor in colorectal cancer". This statement should be corrected. The authors show that there is a correlation between Wnt target gene expression and GNAI2 levels, but do not provide functional evidence that Gai2 effects colorectal cancer development/carcinogenesis.

Figure 4H: The authors show that Gai2 mutations impair the Wnt inhibiting effect of Gai2. However, the authors selected HEK cells that do not harbor mutations in the destruction complex. It would be interesting to see if the same effects can be observed when these mutant forms of Gai2 are expressed in DLD1 or SW480.

Figure 4L: Given the small number of patients carrying loss of GNAI2, did the authors check if the two groups are normalized regarding tumor stage (for instance, metastatic versus localized stage?)

Figure 5/6: One major question is how formation of condensates correlates with reduction of Wnt activity (TOP Flash/Axin2 levels) and viability of cancer cell lines when they are treated with GBZ. The Figures 5B and G, although a little different in set-up, suggest that only a fraction of 10% of cells show condensates at 50 μ M of GBZ, but Wnt signaling is almost reduced by approx. 75% at the same drug dose. How do the authors explain this difference?

Figure 8H: it is confusing for the reader that SW480 is not presented as labelled. The authors should consider shifting the data for DLD1 to a separate (supplementary) figure.

Figure 8: The authors used the APC min model, but only measured tumor size, whereas survival time of the animals and time till occurrence of rectal bleeding a more common parameters to assess the tumor burden. Do the authors have data on these parameters?

Do the authors also observe a downregulation of Wnt target genes in the engrafted tumors treated with guanabenz?

The authors should mention in the discussion that the doses of guanabenz needed to reduce Wnt signaling are very high, raising the question if the same effects can be achieved in vivo with more physiological drug concentrations.

The authors should also more extensively cite and mention literature on the roles of G-proteins in Wnt signaling.

Reviewer #3 (Remarks to the Author):

In this study, the authors uncover Gai2 as a novel Conductin (Cdt; or Axin2) binding partner that promotes its suppressor activity. In their previous work, the authors described an aggregation prone stretch (aggregon) in the Cdt-RGS domain, which promotes RGS-RGS aggregation and prevents the formation of Cdt condensates that are required for beta-catenin destruction. Here, Gai2 was found to act as a steric competitor for Cdt RGS aggregation, promote the formation of Cdt condensates and enhance beta-catenin degradation. Importantly, Gai2 suppresses Wnt/beta-catenin signaling in APC-mutant colon cancer cell lines. Furthermore, using an FDA-approved agonist of the α_2 -receptors, GBZ, the authors show endogenous stimulation of Gai2 can reduce growth of colon cancer cell lines and intestinal organoids in a Cdt-dependent manner. Lastly, in a mouse xenograft model using an APC-mutant colon cancer cell line, GBZ was shown to reduce tumor size.

While the presented concepts are novel and of potential interest, the presentation of results is not clear and transparent throughout the manuscript. Part of the data are presented in a complex and/or indirect manner, or appear overinterpreted, making it difficult to judge robustness of the data. In particular, a role of Gai2-mediated regulation of PKA signaling has not been excluded convincingly. Also, control experiments are missing at a few places and methods need better description.

Major points:

1) Methods for quantification of cells with condensates needs much better specification. What are the criteria used for scoring? Currently, the interpretation of results seems rather arbitrary, as some of the images that were classified as lacking condensates (e.g. Fig. 1C, M3 mutant or Fig. 2E, QVL mutant) are very similar to those with condensates (e.g. Fig. 1C, lower example of +Gai2).

2) The dual role of Gai2 in regulating PKA signaling and Cdt condensation confounds interpretation of the data at a few places. In Fig. 3H-L, Fig. S6D, the expression of Gai2 clearly suppresses Wnt-mediated responses in Cdt^{-/-} cells, likely due to PKA inhibition. Can the authors confirm that the effects of H89 treatment cannot be rescued by Cdt deletion in SW480 and DLD1 cells?

Furthermore, the authors employ AIF4 treatment (the molecular basis of which is not well explained) to increase the affinity of Gai2 with the Axin RGS domain and promote Cdt condensation (Figs 2 and 3), without affecting PKA activity. As 30 μ M AIF4 show similar levels of condensation as 50 μ M GBZ, the authors may use these treatment conditions to correlate Cdt condensation to Wnt pathway and tumor growth inhibition in the absence of altered PKA signaling levels.

3) While Gai3 also binds the CdtRGS domain and promotes Cdt condensate formation (e.g. Fig 2D, Fig S2), it does not seem to rescue Gai2 kd in various model systems. How do the authors explain

these findings? Is Gai3 functionally redundant to Gai2, and does Gai3 expression also correlate with Wnt target gene expression and poor prognosis in cancer subsets?

4) Fig 5: Very high GBZ concentrations (50uM) are applied to demonstrate functional effects, which likely increases the risk for off-target effects. Since GBZ is an FDA-approved drug, how do these concentrations relate to applications in human patients? What is known about toxicity and off-target effects? As mentioned by the authors, GBZ mediates activation of α 2-receptors that in turn activate Gai signaling. How do these molecular activities link to enhanced Gai2 binding to Cdt? Are Gai2 levels increased? How do cellular responses depend on α 2-receptor levels and activity? What is the effect of 50uM GBZ treatment on PKA activity?

Some data are unclear or appear overinterpreted:

5) For instance, the authors indicate a 'much stronger Wnt pathway inhibition by Gai2 as compared to H-89' (Fig. S7C). However, the difference appears 2-fold versus 5-fold, indicating that the effect of H-89 is not negligible. This statement needs to be rephrased.

6) P12: "Moreover, with the aggregating conductin RGS domain, we reveal an example for an interaction site that hinders phase separation, challenging the model of phase separation as result of additive interactions". This statement does not appear justified by the data shown. It is fine to add a new view, e.g. the presence of interaction sites that counteract/balance phase separation. I do not see, however, how the presented data 'challenge' the wealth of LLPS data out in the literature.

7) Fig. 6E-G: The description of these experiments is rather obscure, and representation of the data is indirect ('growth reduction'). A display of absolute growth would be more transparent here, as Cdt deletion itself may already show growth rate effects.

8) Conclusion: 'Thus, endogenous Gai2 suppresses Wnt signaling in colorectal cancer, indicating that the observed decrease of Gai2 expression in tumors drives carcinogenesis by allowing higher Wnt pathway activity.' This statement is not substantiated by the data that remain merely correlative for human colorectal cancer. As stated by the authors, Gai2-deficient mice develop adenocarcinoma in the colon, but this is preceded by colitis. The authors uncover a link between GNAI2 alterations that are merely correlative at this point for colorectal cancer in general.

Other points:

1) Fig. S4: Information on how the novel aggregation site was designed is lacking.

2) Fig 2E+F: The quality of the western blot shown appears not sufficient for quantification. In addition, it is not clear why inputs levels of denaturing samples are equal while they are different for the samples on native gel. Equal total levels should be shown (similar to Fig S4D), then quantification would not even be necessary. If conditions in 2E are combined with AIF4 treatment, are higher order complexes further decreased?

3) Fig. 2K: a representative GST-pulldown blot should be included. Interaction of Gai2 are shown only for Cdt-QVL, which may represent a misfolded protein. Does the Cdt-QV mutant also lose interaction with Gai2?

4) Western blots are missing for multiple experiments (e.g. Fig S5, Fig S6)

5) Supplier information and refs for GBZ are missing

REVIEWER COMMENTS

We thank all three reviewers for their constructive questions and suggestions, which we address point by point as follows:

Reviewer #1 (Remarks to the Author):

Miete and colleagues describe a novel mechanism that regulates the function of Axin2 (Conductin) as a suppressor of b-catenin signaling through heterotrimeric G proteins. The premise is that G α i2 (and to a lesser extent of the G α ihai proteins) bind to the RGS domain of Axin2 to allow its aggregation in cytoplasmic puncta and also to favor its function as a suppressor of b-catenin signaling. One of the strengths of the work is the identification of an FDA-approved agonist (GBZ) of GPCRs that activate G proteins as a potential anti-colorectal cancer treatment that presumably operates through the G α i2-Axin2 mechanism proposed here. This last part of the manuscript on the action of GBZ is convincing and potentially impactful. The area that, in my opinion, would need significant improvement is on the molecular mechanism of G α i-mediated regulation of Axin2. The following points are listed in approximate order of importance.

1- The authors need to provide more definitive data on the requirement for G α i2-Axin2 binding for the regulation of Axin2 localization and function. Evidence so far is correlative and to establish definitely the mechanism the authors will need a mutant that specifically disrupt the G α i2-Axin2 interaction. For clarity, the use of ALF4- gives indirect evidence and might have many indirect consequences, and the use of Axin2 mutants with swapped aggregation positions does not directly address the need for G α i2 binding.

To address this suggestion, we tested two different G α protein mutations that had been associated with reduced RGS binding (G42R and G184S) for whether they attenuate G α i2-RGS^{Cdt} interaction. In contrast to G42R, which had no effect, the G184S mutation reduced binding to the conductin RGS domain by about 50% (new Fig. 2c and Supplementary Fig. 4b). (Binding of G α i2 to RGS19, which we investigated in parallel as control, was more strongly impaired by the G184S mutation, as shown in the image for the reviewer below.) In line with reduced conductin binding, the G184S mutant was less efficient in inducing polymerization of conductin (new Fig. 2d), and notably also in inhibiting Wnt signaling (new Supplementary Fig. 7f, g), as compared to WT G α i2.

2- The interaction between Gai2 and Axin2 needs to be characterized more thoroughly, including experiments with purified components rather than using cellular expression. One specific point is the characterization of Axin2-RGS binding as function of the activation status of Gai2. The authors imply that the interaction occurs with the active Gai2, but this is at odds with results presented in Figure 2A, B, in which cellular Gai2 will be predominantly inactive. Loading with GDP, GTP γ S and AIF4- should be compared.

We investigated the G α i2-conductin interaction using purified recombinant proteins, as suggested. We found that the isolated conductin RGS domain (and an N-terminal conductin fragment containing the RGS domain, Cdt 2-210) bound significantly stronger to active G α i2 preloaded with GTP γ S or GDP+AIF $_4^-$ compared to inactive GDP-loaded G α i2 (new Fig. 2f and Supplementary Fig. 4c). We think that higher affinity for active G α i2 is not at odds with Fig. 2a, b, as (i) some G α i2 will be in the active state in this experimental setup and (ii) we observed binding also with inactive G α i2 to some extent.

3- If Axin2 binds to active Gai2 or to the AIF4-bound mimic of the active transition state, it might be acting as a GAP. This should be evaluated.

We evaluated a potential GAP function of the conductin RGS domain using the fluorescent GTP derivative BODIPY-FL-GTP. Neither the isolated conductin RGS domain (His- or GST-tagged) nor a N-terminal conductin fragment containing the RGS domain accelerated BODIPY-FL-GTP hydrolysis by G α i2, which was measured via the decrease of fluorescence intensity, suggesting that the conductin RGS domain has no GAP function (new Supplementary Fig. 4d, e). RGS10 was used as positive control. Preferential binding to the active state of G α -proteins without GAP function has already been described for other RGS domain containing proteins including axin^{1, 2}.

4- The authors should expand the G protein subtypes investigated. They have only investigated members of the Gi/o family, but Gs, G12 or Gq family G α subunits may also bind to Axin2.

We expanded the analysis of G protein subtypes by investigating G α s, G α 12 and G α q, as suggested. While G α s and G α q were markedly less active in triggering conductin polymerization compared to G α i2, G α 12 potently induced polymerization (new Supplementary Fig. 3a-c), and may represent an interesting candidate for future studies.

5- A significant limitation of the mechanistic studies is the overreliance on overexpression systems. There are no experiments to document that endogenous Axin2 behaves like the overexpressed one. Key experiments should be performed to demonstrate that endogenous Axin2 shifts distribution upon manipulation of endogenous Gai2.

Unfortunately, we were not able to stain endogenous conductin/axin2 in immunofluorescence experiments most likely because of the low expression levels³.

However, we observed a dosage-dependent downshift of the complexes formed by endogenous conductin upon GBZ treatment in native gels (new Fig. 6c), suggesting that activation of endogenous G α i2 by GBZ reduces aggregation of endogenous conductin, similar as observed for overexpressed conductin (Fig. 2i-k).

6- Claims implying that Axin2 distributes to phase separated condensates and that phase separation is important for function should be omitted. The evidence demonstrating that Axin2 is in phase separated condensates is weak for the overexpressed protein and absent for endogenous one, ...

On the one side, we agree that our manuscript lacks in depth biophysical characterization of conductin polymers as phase separating biomolecular condensates. That is why we toned down conclusions on conductin phase separation, as follows. "We characterize conductin polymers as

biomolecular condensates by proving key condensate properties for conductin puncta” was changed to “we observed key properties of biomolecular condensates for conductin puncta”, “with high functional relevance” was deleted from the next sentence and the following sentence about condensates with therapeutic potential was deleted completely. In addition, “challenging the model of phase separation” was changed to “adding a new aspect to the phase separation model”, following a suggestion of reviewer #3. (Changes were made in the discussion p. 13).

On the other side, a very recent publication characterized axin as phase separating protein in detail⁴, and we assume that this is also the case for conductin, as both homologs polymerize via a conserved DIX domain-dependent mechanism. Consistently, we did observe condensate characteristics for the conductin polymers, such as shape, assembly dynamic, fusion, temperature sensitivity and concentration dependency. Therefore, we prefer to abstain from deleting “condensate/condensation” throughout the manuscript, also as the other reviewers had no principal objections to use the term. The new information about axin was added to the introduction (p. 3).

... and the evidence to support a causal relationship between phase separation and function is non-existent. There are only correlations between subcellular localization and function.

We established a conductin mutant (QVL) that did not form condensates upon $G\alpha i2$ binding (Fig. 3 / old Fig. 2 G-L). Of note, $G\alpha i2$ did not promote β -catenin degradation by this QVL mutant, suggesting a causal relationship between cellular distribution and function (Fig. 4a, b /old Fig. 3 A, B). In line with this, earlier studies demonstrated a functional relationship between polymerization of axin proteins and β -catenin degradation^{5, 6, 7, 8}.

7- Figure 3F, a control showing that Wnt activity can be further increased is missing.

We included a respective control demonstrating that Wnt activity can be further increased in siCdt transfected U2OS cells by e.g. Wnt3a treatment (new Supplementary Fig. 7c, d).

8- What is the impact of the $G\alpha i2$ mutants described in Figure 4 on Axin2 binding and in regulating Axin2 localization?

We found that induction of conductin polymerization was moderately yet significantly reduced for the cancer mutants compared to WT $G\alpha i2$ (new Supplementary Fig. 10c). In binding assays, which were performed similarly as in Fig. 2a, we did not observe a convincing decrease in binding. We feel that the reporter assay is the most sensitive and quantitative assay of the three, and differences in binding or regulating distribution are harder to reveal.

9- I suggest removing "severe" when referring to the differences in patient survival described in the manuscript. The differences are barely statistically significant. This is in part due to the low frequency of genomic $G\alpha i2$ alterations. The low prevalence of these alterations should also be explicitly discussed (ie, this might not be a driver event like other alterations of the pathway are).

We changed the sentence from “... whose inactivation has severe consequences for patient survival.” to “...whose inactivation is associated with reduced patient survival.”

To compare the frequency of $G\alpha i2$ alterations to oncogenic driver mutations of other Wnt pathway components, the following sentence was added to the discussion: “Inactivating *GNAI2* alterations occur in about 6% of the colorectal cancer patients, which is about the frequency of oncogenic β -catenin (5%) or conductin (7%) mutations but far below loss of APC (77%), the major tumor suppressor of the Wnt pathway⁹.”

Reviewer #2 (Remarks to the Author):

In the work by Miete et al. the authors show that Gai2 induces condensation of AXIN2/conductin, which is mediated by targeting an aggregon in the RGS domains and leads to the reduction of Wnt signaling. Furthermore, the authors demonstrate the levels of Gai2 are frequently reduced in CRC. Using the Gai2 activating drug guanabenz, the authors use several models of colorectal cancer to show that this drug can diminish Wnt levels and reduce growth of CRC. While the molecular aspect of Gai2/conductin interaction is convincing, the cancer-related focus of the manuscript needs revision. Specifically, it is unclear to which extent guanabenz works through Gai2 to reduce growth of CRC cell lines.

Overall, this is an interesting manuscript building on the e 2019 Nat Comm publication by Bernkopf et al. and providing mechanistic insights into aggregon regulation of axin2/conductin through Gai2 and condensate regulation in general.

Major comments:

As the authors suggest that antiproliferative effects of GBZ functions (partly) through Gai2, the question is if overexpression of Gai2 can phenocopy the effect of GBZ and if siGai2 can rescue tumor cells from the antiproliferative effects of GBZ? The authors should provide additional evidence to show that GBZ works through Gai2.

As requested, we now do provide additional evidence that GBZ works through Gai2. Indeed, Gai2 knockdown significantly rescued GBZ-induced growth inhibition of SW480 and DLD1 cancer cells (new Fig. 7h, i and Supplementary Fig. 13d, e), and overexpression of Gai2 phenocopied growth inhibition in both cell lines (new Fig 7j and Supplementary Fig. 13f), consistent with our previous data.

[Please note that we had to renumber some figures due to the new data, and old Fig. 6 was split into new figures 7 and 8. The DLD1 data was moved to the Supplementary Fig. 13.]

Minor comments:

Figure 1F: could be transferred to the supplement

We prefer to keep Fig. 1f because it contains relevant information and allows a compact figure arrangement, without distracting from major points.

Figure 2B: In this figure, it is unclear why Gai2 is only compared to Gai0 and not to the other Gai

There was indeed no specific reason for this, and we added the suggested statistical comparisons of Gai2 to Gai1 and to Gai3.

Figure 2L: since the approach is not easy to understand, the model in L should be presented before H-K

As suggested, we moved Fig. 2 L and explained our approach/hypothesis before starting with the aggregon swapping experiments. Please note that we had to split old Fig. 2 due to newly included data, and Fig. 2 L is now Fig. 3a.

“Wnt/ β -catenin signaling by promoting conductin condensation in a broad variety of different tissues including colorectal cancer” > this sentence should be rephrased as the authors showed the phenotype in 3-4 different cell lines from 3 tissue backgrounds, and “a broad variety of different tissues” seems inappropriate.

The sentence was changed to “in a variety of different cell lines including colorectal cancer cells”.

Figure 3H and I: The normalization of *-/siCdt* seems odd in these sub-figures. As shown in 3F, *siCtrl* activates TOP Flash reporter activity in a strong fashion, but the way the data is presented in H and I, it seems that there is no difference between the two conditions

We appreciate the careful analysis of our data by the reviewer. Indeed, *conductin* knockdown also increased Wnt signaling activity in the experiments Fig. 3 H and I. Without normalization of *-/siCdt*, all *siCdt* bars would be higher, so that the *Gai2* effects in control vs *siCdt* cells would be hard to compare, and the reader would have to calculate himself considering the different starting activities. That is why we believe that the chosen approach of visualization allows the best judgement of the *Gai2* effects in control cells vs *siCdt* cells. To exclude any confusion of the reader about the *siCdt* effect in Fig. 3H and I (new Fig. 4h, i), we added a sentence to the respective figure legends explaining the normalization strategy.

Figure 3K: it is not clear in this figure if knockout of *conductin* increases basal Wnt levels. The current presentation of the data suggests that there is no difference. Also labelling as *-/-* is confusing, the use of *sgCdt* or *Cdt KO* would be easier to comprehend.

Yes, knockout of *conductin* activates Wnt signaling in SW480 cells (see Fig. 4 E [old manuscript] or Fig. 5e [new manuscript]). For the explanation why we have chosen the way of presentation in Fig. 3K, please see above.

As suggested, *-/-* was changed to *Cdt KO* throughout the manuscript.

“*Gai2* functions as tumor suppressor in colorectal cancer”. This statement should be corrected. The authors show that there is a correlation between Wnt target gene expression and *GNAI2* levels, but do not provide functional evidence that *Gai2* effects colorectal cancer development/carcinogenesis.

The statement was changed to “*Gai2* aberrations are associated with reduced patient survival”.

Figure 4H: The authors show that *Gai2* mutations impair the Wnt inhibiting effect of *Gai2*. However, the authors selected HEK cells that do not harbor mutations in the destruction complex. It would be interesting to see if the same effects can be observed when these mutant forms of *Gai2* are expressed in DLD1 or SW480.

As suggested, we analyzed the *Gai2* cancer mutations in DLD1 cells. Similar as observed in HEK293T cells, the three *Gai2* mutants inhibited Wnt signaling significantly less compared to WT *Gai2*, which was irrespective of expression levels (new Supplementary Fig. 10a, b). Interestingly, the K210N mutation had the weakest effect in HEK293T cells but the strongest effect in DLD1 cells, while the T328S mutation behaved just the opposite, suggesting that the effect intensity of the cancer mutations is cell context specific. Here, the mentioned mutations in the destruction complex and/or the *conductin* expression levels may play a role, as they are much higher in DLD1 cells compared to HEK293T cells.

Figure 4L: Given the small number of patients carrying loss of *GNAI2*, did the authors check if the two groups are normalized regarding tumor stage (for instance, metastatic versus localized stage?)

Our data show that *GNAI2* loss activates Wnt signaling, which is known to drive colorectal cancer progression. Thus, we actually assume more advanced tumor stages in consequence of *GNAI2* loss in this group, and normalization regarding tumor stage would obscure the effect of *GNAI2* loss. That is why we did not normalize the two groups for tumor stages.

Figure 5/6: One major question is how formation of condensates correlates with reduction of Wnt activity (TOP Flash/Axin2 levels) and viability of cancer cell lines when they are treated with GBZ. The Figures 5B and G, although a little different in set-up, suggest that only a fraction of 10% of cells show condensates at 50 μ M of GBZ, but Wnt signaling is almost reduced by approx. 75% at the same drug dose. How do the authors explain this difference?

Fig. 5 B and G differ in the experimental setup, and thus it is hard to compare the GBZ activity in both assays quantitatively. In Fig. 5 B, conductin was transiently overexpressed to visualize condensate induction by GBZ because endogenous conductin levels are rather low and difficult to stain, while there were only the endogenous expression levels of conductin in Fig. 5 G. Since GBZ functions via the limited endogenous $G\alpha i2$ pool, it is well possible that the higher conductin levels in Fig. 5 B are less efficiently affected compared to the lower, endogenous conductin levels in Fig. 5 G. In addition, we speculate that GBZ did induce condensates in most of the cells, which however remain too small to become microscopically visible in 90% of the cells (Fig. 5 B). These small condensates may also contribute to enhanced β -catenin degradation, as they might still be several folds bigger than aggregates.

Figure 8H: it is confusing for the reader that SW480 is not presented as labelled. The authors should consider shifting the data for DLD1 to a separate (supplementary) figure.

The labelling was changed accordingly, and the data for DLD1 cells was moved to the separate Supplementary Fig. 13c.

Figure 8: The authors used the APC min model, but only measured tumor size, whereas survival time of the animals and time till occurrence of rectal bleeding a more common parameters to assess the tumor burden. Do the authors have data on these parameters?

According to governmental and ethical guidelines on animal welfare, we did not follow up the mice until death to generate survival curves, but sacrificed all animals with about 15 weeks, when the harm to the animals did not exceed moderate levels.

We do not have data on rectal bleeding. However, we do have data on splenomegaly of the mice, which is another indirect parameter positively correlating with the overall tumor burden of APC^{Min} mice^{10, 11, 12}. Of note, GBZ treatment partially rescued splenomegaly, in line with reduced tumorigenesis in these animals (new Supplementary Fig. 15).

Do the authors also observe a downregulation of Wnt target genes in the engrafted tumors treated with guanabenz?

We did quantify mRNA expression of Wnt target genes in the engrafted tumors. However, we did not observe a significant reduction in target gene expression, probably due to the overall high variances between individual tumors of up to 4-fold.

The authors should mention in the discussion that the doses of guanabenz needed to reduce Wnt signaling are very high, raising the question if the same effects can be achieved in vivo with more physiological drug concentrations.

New data showed that inhibition of a luciferase reporter for PKA signaling required similar GBZ concentrations in our hands as for the Wnt signaling reporter (new Supplementary Fig. 12j). Since physiologic GBZ effects such as in hypertension patients are mediated via $G\alpha i$ -PKA-signaling, we are positive that physiologic concentrations will also suffice for Wnt signaling inhibition, as suggested by our mouse studies. In addition, although we do not know the intestinal concentrations of our treated animals, enrichment of GBZ in tissues up to double-digit μ M concentrations was shown previously¹³.

Related to this dose issue, we found that the GBZ amount applied for tumor treatment in our mouse studies translates to a human equivalent dose of 16.8 mg GBZ per day for a 60 kg patient (see new discussion paragraph p. 15 for calculation). This is far below GBZ doses that have been safely used to treat hypertension patients (up to 64 mg per day), and thus repurposing GBZ may be a real option for cancer treatment.

The authors should also more extensively cite and mention literature on the roles of G-proteins in Wnt signaling.

We added a new paragraph to the Introduction about the roles of G-proteins in Wnt signaling.

Reviewer #3 (Remarks to the Author):

In this study, the authors uncover G α i2 as a novel Conductin (Cdt; or Axin2) binding partner that promotes its suppressor activity. In their previous work, the authors described an aggregation prone stretch (aggregon) in the Cdt-RGS domain, which promotes RGS-RGS aggregation and prevents the formation of Cdt condensates that are required for beta-catenin destruction. Here, G α i2 was found to act as a steric competitor for Cdt RGS aggregation, promote the formation of Cdt condensates and enhance beta-catenin degradation. Importantly, G α i2 suppresses Wnt/beta-catenin signaling in APC-mutant colon cancer cell lines. Furthermore, using an FDA-approved agonist of the α 2-receptors, GBZ, the authors show endogenous stimulation of G α i2 can reduce growth of colon cancer cell lines and intestinal organoids in a Cdt-dependent manner. Lastly, in a mouse xenograft model using an APC-mutant colon cancer cell line, GBZ was shown to reduce tumor size.

While the presented concepts are novel and of potential interest, the presentation of results is not clear and transparent throughout the manuscript. Part of the data are presented in a complex and/or indirect manner, or appear overinterpreted, making it difficult to judge robustness of the data. In particular, a role of G α i2-mediated regulation of PKA signaling has not been excluded convincingly. Also, control experiments are missing at a few places and methods need better description.

We thank the reviewer for her/his interest in our work, and the overall positive assessment of the manuscript. We identified a novel G α i2-conductin dependent mechanism to inhibit Wnt signaling via conductin condensation, which accounts for at least 50% of the Wnt pathway inhibition by G α i2, according to our combined data. However, we do not exclude a contribution via G α i2-mediated regulation of PKA signaling, which was already appreciated in the initial submission (see e.g. p. 7). Actually, we think that this dual-impact mechanism points to a special physiological relevance (see first discussion paragraph).

What we exclude is a role for PKA signaling in regulating condensation of conductin (see paragraph "Conductin polymerizes independently of G α i2-PKA signaling").

Major points:

1) Methods for quantification of cells with condensates needs much better specification. What are the criteria used for scoring? Currently, the interpretation of results seems rather arbitrary, as some of the images that were classified as lacking condensates (e.g. Fig. 1C, M3 mutant or Fig. 2E, QVL mutant) are very similar to those with condensates (e.g. Fig. 1C, lower example of +G α i2).

The three mentioned images, i.e. Fig. 1C lower example of +G α i2, Fig. 1C M3 mutant and Fig. 2E QVL mutant (new Fig. 3c) are all meant to show membrane recruitment (arrowheads), and are therefore indeed similar, as judged by the reviewer. Only the upper example of +G α i2 in Fig. 1C is meant to show condensates. We regret this misunderstanding, and we added a sentence to the legend of Fig. 1 for clarification.

In addition, we added a sentence describing the classification to the immunofluorescence section (Methods).

2) The dual role of G α i2 in regulating PKA signaling and Cdt condensation confounds interpretation of the data at a few places. In Fig. 3H-L, Fig. S6D, the expression of G α i2 clearly suppresses Wnt-mediated responses in Cdt^{-/-} cells, likely due to PKA inhibition.

We agree, and this interpretation was already mentioned in the previously submitted manuscript p. 7 (“The residual $G\alpha i2$ activity in conductin knockout cells (Fig. 3 K) most likely depends on PKA inhibition,...”)

Can the authors confirm that the effects of H89 treatment cannot be rescued by Cdt deletion in SW480 and DLD1 cells?

We found that H-89 inhibited Wnt signaling to a similar extent in SW480 WT and conductin knockout cells (new Supplementary Fig. 8e), demonstrating that inhibition of Wnt signaling via PKA inhibition is independent of conductin. We think that this clear distinction between a conductin-dependent (conductin polymerization) and a conductin-independent (PKA inhibition) mechanism to inhibit Wnt signaling by $G\alpha i2$ increases the specificity of our findings, and we thank the reviewer for this supportive suggestion.

Furthermore, the authors employ AIF4 treatment (the molecular basis of which is not well explained) to increase the affinity of $G\alpha i2$ with the Axin RGS domain and promote Cdt condensation (Figs 2 and 3), without affecting PKA activity. As 30 μ M AIF4 show similar levels of condensation as 50 μ M GBZ, the authors may use these treatment conditions to correlate Cdt condensation to Wnt pathway and tumor growth inhibition in the absence of altered PKA signaling levels.

We apologize for the insufficient explanation of the AlF_4^- mechanism, and we added a respective sentence to the Results section. Basically, AlF_4^- binds to inactive GDP-loaded $G\alpha$ -proteins within the GDP-binding pocket, thereby mimicking the active GTP-loaded state. AlF_4^- treatment has pleiotropic effects as it may activate all kinds of $G\alpha$ proteins, and there is experimental evidence that AlF_4^- -activated $G\alpha$ proteins regulate PKA activity^{14, 15}. Thus, the assumption of the reviewer that AlF_4^- treatment will not affect PKA activity is not correct, and we can therefore not use AlF_4^- treatment instead of GBZ to correlate Cdt condensation to Wnt pathway and tumor growth inhibition.

3) While $G\alpha i3$ also binds the CdtRGS domain and promotes Cdt condensate formation (e.g. Fig 2D, Fig S2), it does not seem to rescue $G\alpha i2$ kd in various model systems. How do the authors explain these findings?

$G\alpha i3$ is less efficient compared to $G\alpha i2$ in promoting condensation and conductin binding (Fig. 1d, 2b). In addition, $G\alpha i3$ is less expressed compared to $G\alpha i2$ in e.g. U2OS or HEK293 cells (Human Protein Atlas available from <http://www.proteinatlas.org>)¹⁶. Lower binding affinity and lower expression are probably the reasons why $G\alpha i3$ cannot compensate for loss of $G\alpha i2$ activity in the knockdown experiments.

Is $G\alpha i3$ functionally redundant to $G\alpha i2$, and does $G\alpha i3$ expression also correlate with Wnt target gene expression and poor prognosis in cancer subsets?

Particularly in the colon, expression of $G\alpha i3$ is more than 40x lower as compared to $G\alpha i2$ (about 3,5 transcripts per million vs about 160; GEPIA analysis as described in the manuscript)¹⁷. Even in colorectal cancers, which show decreased $G\alpha i2$ expression compared to healthy colon (Fig. 4 C [old manuscript], Fig. 5c [new manuscript]), expression of $G\alpha i2$ is still about 20x higher compared to $G\alpha i3$. Consistent with the very low expression levels, we observed no correlation between mRNA expression of $G\alpha i3$ and β -catenin target genes *LGR5* ($R = -0.006$, compared to -0.51 with $G\alpha i2$), *AXIN2* ($R = -0.12$, compared to -0.41 with $G\alpha i2$), *RNF43* ($R = -0.17$, compared to -0.75 with $G\alpha i2$) and *ASCL2* ($R = -0.084$, compared to -0.50 with $G\alpha i2$). Therefore, we believe that $G\alpha i3$ does not function redundantly to $G\alpha i2$ in intestinal Wnt signaling inhibition and during colorectal carcinogenesis.

4) Fig 5: Very high GBZ concentrations (50 μ M) are applied to demonstrate functional effects, which likely increases the risk for off-target effects. Since GBZ is an FDA-approved drug, how do these concentrations relate to applications in human patients? What is known about toxicity and off-target effects?

We calculated that 50 μ M GBZ in the drinking water of the mice resulted in a daily uptake of about 3.47 mg/kg GBZ, which translates to a human equivalent dose of about 0.28 mg/kg per day, using the FDA recommended conversion based on body surface area. This corresponds to about 16.8 mg GBZ per day for a 60 kg cancer patient. As hypertension patients were safely treated with up to 64 mg GBZ per day, we do not expect toxicity or off-target effects, and repurposing GBZ may offer a feasible chance for cancer treatment. For calculation details and references, please refer to the new paragraph of the discussion p. 15 explaining this issue.

As mentioned by the authors, GBZ mediates activation of α 2-receptors that in turn activate G α i signaling. How do these molecular activities link to enhanced G α i2 binding to Cdt? Are G α i2 levels increased?

We assume that activation of G α i2 follows the standard way of trimeric G-protein signaling: In the inactive state, G α i2 is loaded with GDP and bound in a trimeric complex together with a G β and a G γ subunit, and this complex is coupled to the α 2-adrenoceptor. Activation of the receptor promotes GDP to GTP exchange in G α i2 and dissociation of the G $\beta\gamma$ -subunits. Thus, the active GTP-loaded G α i2 is now free to interact with conductin. We do not think that G α i2 levels are increased in addition, and we have no experimental evidence for increased G α i2 expression after GBZ treatment.

How do cellular responses depend on α 2-receptor levels and activity?

To answer this question, we performed knockdown experiments. Of note, α 2a-adrenoceptor knockdown markedly impaired Wnt signaling inhibition by GBZ, showing dependency on α 2-adrenoceptor levels (new Supplementary Fig. 12h, i).

What is the effect of 50 μ M GBZ treatment on PKA activity?

We investigated the effect of GBZ on PKA activity in CRE luciferase reporter assays. We found that 50 μ M GBZ inhibited PKA activity by about 45% in HEK293T cells (new Supplementary Fig. 12j), while Wnt signaling was inhibited by about 90% by 50 μ M GBZ (Supplementary Fig. 12a).

Some data are unclear or appear overinterpreted:

5) For instance, the authors indicate a 'much stronger Wnt pathway inhibition by G α i2 as compared to H-89' (Fig. S7C). However, the difference appears 2-fold versus 5-fold, indicating that the effect of H-89 is not negligible. This statement needs to be rephrased.

The statement was rephrased to "... about twofold stronger Wnt pathway inhibition by G α i2 compared to specific PKA inhibition by H-89 ...".

6) P12: "Moreover, with the aggregating conductin RGS domain, we reveal an example for an interaction site that hinders phase separation, challenging the model of phase separation as result of additive interactions". This statement does not appear justified by the data shown. It is fine to add a new view, e.g. the presence of interaction sites that counteract/balance phase separation. I do not see, however, how the presented data 'challenge' the wealth of LLPS data out in the literature.

We agree and changed the sentence to "... adding a new aspect to the phase separation model.", as suggested.

7) Fig. 6E-G: The description of these experiments is rather obscure, and representation of the data is indirect ('growth reduction'). A display of absolute growth would be more transparent here, as Cdt deletion itself may already show growth rate effects.

We want to point out that Fig. 6 E (= Fig. 7e in the revised manuscript) shows the requested absolute growth for SW480 WT and Cdt knockout cells revealed by MTT color intensity (= number of viable cells). Here, raw values without further calculations are shown. For the DLD1 cells, we now included a similar figure (new Supplementary Fig. 13a).

As becomes apparent from Fig. 7e and the new Supplementary Fig. 13a, conductin knockout or knockdown alone has (if at all) very minor growth rate effects.

[Please note that the DLD1 data on cell growth and cell division were moved to the new Supplementary Fig. 13, following a suggestion of reviewer #2]

For clarification, we optimized the description of how "growth reduction" was calculated in the figure legend.

8) Conclusion: 'Thus, endogenous G α i2 suppresses Wnt signaling in colorectal cancer, indicating that the observed decrease of G α i2 expression in tumors drives carcinogenesis by allowing higher Wnt pathway activity.' This statement is not substantiated by the data that remain merely correlative for human colorectal cancer. As stated by the authors, G α i2-deficient mice develop adenocarcinoma in the colon, but this is preceded by colitis. The authors uncover a link between GNAI2 alterations that are merely correlative at this point for colorectal cancer in general.

We changed the respective sentence to "As knockdown of endogenous G α i2 activated Wnt signaling in colorectal cancer cells, the observed decrease of G α i2 expression in human tumors may likewise allow higher Wnt pathway activity, which promotes carcinogenesis."

Other points:

1) Fig. S4: Information on how the novel aggregation site was designed is lacking.

We added a respective sentence to the figure legend, explaining how the idea to activate a novel aggregation site by the L99R mutation developed. Please note that old Fig. S4 is now Supplementary Fig. 5.

2) Fig 2E+F: The quality of the western blot shown appears not sufficient for quantification. In addition, it is not clear why inputs levels of denaturing samples are equal while they are different for the samples on native gel. Equal total levels should be shown (similar to Fig S4D), then quantification would not even be necessary. If conditions in 2E are combined with AIF4 treatment, are higher order complexes further decreased?

We added an alternative blot with rather equal total levels to the supplement, which now also includes G α o, clearly illustrating G α i2-induced decrease and increase of large and small molecular weight complexes, respectively (new Supplementary Fig. 4j). Unfortunately, we were not able to improve the overall native blot quality, as the major problem for better detection was the spreading of HA-Cdt 2-345 over a blot area that is rather big compared to a concrete band. We agree that the total protein levels of the native blot in Fig. 2 E (Fig. 2i revised manuscript) do not appear equal. However, the quantification is normalized for such differences because the abundance of large, medium and small complexes is presented relatively as percentage of the total protein amount of the respective track.

In addition, we investigated the effect of AIF₄⁻ treatment, as suggested. AIF₄⁻ treatment without G α i2 co-expression was sufficient to decrease aggregation of HA-Cdt 2-345 in a dosage-dependent

manner, as revealed by a downshift of especially the medium size complexes (new Fig. 2k), which is consistent with induction of conductin condensation by AlF_4^- (old Fig. S3 D / new Supplementary Fig. 4h). The combination of $G\alpha i2$ expression and AlF_4^- treatment did not reproducibly enhance the $G\alpha i2$ effect (data not shown), however the effect of combining both in immunofluorescence experiments was also not enormous (old Fig. S3 B / new Supplementary Fig. 4f)

Moreover, following a suggestion by reviewer #1, we found that GBZ treatment reduced aggregation of endogenous conductin (new Fig. 6c). Thus, altogether, our newly provided data strongly support our original finding that $G\alpha i2$ reduces conductin aggregation, which can be detected on native gels.

3) Fig. 2K: a representative GST-pulldown blot should be included.

The requested GST-pulldown blot was included (new Fig. 3f).

[Please note that we had to renumber some figures due to newly included data.]

Interaction of $G\alpha i2$ are shown only for Cdt-QVL, which may represent a misfolded protein.

Cdt-QVL still bound to $G\alpha i2$ indicating that the folding is rather intact (Fig. 3f, g [Fig. 2 K old manuscript]).

Does the Cdt-QV mutant also lose interaction with $G\alpha i2$?

Notably, the Cdt-QVL mutant did not lose interaction (Fig. 3f, g [Fig. 2 K old manuscript]), as implied by the question. We assume that the Cdt-QV mutant interacts with $G\alpha i2$ similarly as the Cdt-QVL mutant, which also contains the QV mutation.

4) Western blots are missing for multiple experiments (e.g. Fig S5, Fig S6)

We added Western blots to old Fig. S5, showing that forskolin (panels e-h) and Bt_2cAMP treatment (panels j, k, m) do not alter $G\alpha i2$ expression in U2OS and HEK293T cells (new Supplementary Fig. 6i and l). [Panels a-d are based on immunofluorescence experiments, and expression of the proteins was verified via the fluorescence signal.] We also added Western blots to old Fig. S6 showing dosage-dependent expression of $G\alpha i2$ in U2OS and HEK293T cells (new Supplementary Fig. 7e, g) to support Fig. 4g, h and Supplementary Fig. 7f, h. Moreover, we included Western Blots for the newly generated Supplementary Fig. 3a, c, Supplementary Fig. 7c, d (showing the efficiency of siRNA mediated conductin knockdown in U2OS cells) and Supplementary Fig. 10a, b.

5) Supplier information and refs for GBZ are missing

Supplier information for all small molecules (including guanabenz [GBZ]) are provided in the methods section (paragraph: cell culture, transfection and treatment). We now added the order numbers for specification. In addition, we added the reference of the first guanabenz study.

1. Carman CV, *et al.* Selective regulation of $G\alpha i2$ by an RGS domain in the G protein-coupled receptor kinase, GRK2. *J Biol Chem* **274**, 34483-34492 (1999).
2. Stemmler LN, Fields TA, Casey PJ. The regulator of G protein signaling domain of axin selectively interacts with $G\alpha i2$ but not $G\alpha i3$. *Molecular pharmacology* **70**, 1461-1468 (2006).

3. Bernkopf DB, Hadjihannas MV, Behrens J. Negative-feedback regulation of the Wnt pathway by conductin/axin2 involves insensitivity to upstream signalling. *J Cell Sci* **128**, 33-39 (2015).
4. Nong J, Kang K, Shi Q, Zhu X, Tao Q, Chen YG. Phase separation of Axin organizes the beta-catenin destruction complex. *J Cell Biol* **220**, (2021).
5. Fiedler M, Mendoza-Topaz C, Rutherford TJ, Mieszczanek J, Bienz M. Dishevelled interacts with the DIX domain polymerization interface of Axin to interfere with its function in down-regulating beta-catenin. *Proc Natl Acad Sci U S A* **108**, 1937-1942 (2011).
6. Bernkopf DB, Bruckner M, Hadjihannas MV, Behrens J. An aggregon in conductin/axin2 regulates Wnt/beta-catenin signaling and holds potential for cancer therapy. *Nat Commun* **10**, 4251 (2019).
7. Mendoza-Topaz C, Mieszczanek J, Bienz M. The Adenomatous polyposis coli tumour suppressor is essential for Axin complex assembly and function and opposes Axin's interaction with Dishevelled. *Open biology* **1**, 110013 (2011).
8. Pronobis MI, Rusan NM, Peifer M. A novel GSK3-regulated APC:Axin interaction regulates Wnt signaling by driving a catalytic cycle of efficient betacatenin destruction. *Elife* **4**, (2015).
9. Cancer Genome Atlas N. Comprehensive molecular characterization of human colon and rectal cancer. *Nature* **487**, 330-337 (2012).
10. Hodgson A, Wier EM, Fu K, Sun X, Wan F. Ultrasound imaging of splenomegaly as a proxy to monitor colon tumor development in Apc(min716/+) mice. *Cancer Med* **5**, 2469-2476 (2016).
11. You S, *et al.* Developmental abnormalities in multiple proliferative tissues of Apc(Min/+) mice. *Int J Exp Pathol* **87**, 227-236 (2006).
12. Li Q, Lohr CV, Dashwood RH. Activator protein 2alpha suppresses intestinal tumorigenesis in the Apc(min) mouse. *Cancer Lett* **283**, 36-42 (2009).
13. Way SW, *et al.* Pharmaceutical integrated stress response enhancement protects oligodendrocytes and provides a potential multiple sclerosis therapeutic. *Nat Commun* **6**, 6532 (2015).
14. Niu J, *et al.* Interaction of heterotrimeric G13 protein with an A-kinase-anchoring protein 110 (AKAP110) mediates cAMP-independent PKA activation. *Curr Biol* **11**, 1686-1690 (2001).
15. Peeters T, Louwet W, Gelade R, Nauwelaers D, Thevelein JM, Versele M. Kelch-repeat proteins interacting with the Galpha protein Gpa2 bypass adenylate cyclase for direct regulation of protein kinase A in yeast. *Proc Natl Acad Sci U S A* **103**, 13034-13039 (2006).

16. Uhlen M, *et al.* Proteomics. Tissue-based map of the human proteome. *Science* **347**, 1260419 (2015).
17. Tang Z, Li C, Kang B, Gao G, Li C, Zhang Z. GEPIA: a web server for cancer and normal gene expression profiling and interactive analyses. *Nucleic acids research* **45**, W98-W102 (2017).

REVIEWER COMMENTS

Reviewer #1 (Remarks to the Author):

I appreciate the authors' effort to respond to my previous concerns. Many of them have been addressed satisfactorily, but some others remain problematic.

1- Previously I raised the following "to establish definitely the mechanism the authors will need a mutant that specifically disrupt the Gai2-Axin2 interaction." (point 1). In response the authors used a previously described mutant (G184S) that efficiently disrupt binding to RGS GAPs. There are two problems with this: (1) the mutant is not specific for Gai-Axin2, instead it disrupts binding of Gai to many RGS GAPs, so any effect of this mutant could be due to diminished RGS GAP binding, and (2) the effect of this mutation on Axin2 binding is quite modest. Together with their new finding that the interaction is similarly strong with GTPγS and AIF4-loading suggests that Axin2 has an effector-like binding pose rather than GAP-like binding.

Thus, the authors should put effort on designing a mutant of Gai2 or Axin2 that specifically disrupt their interaction without affecting other binders.

2- In response to my previous point 6 ("the evidence to support a causal relationship between phase separation and function is nonexistent."), the authors state that "We established a conductin mutant (QVL) that did not form condensates upon Gai2 binding (Fig. 3 / old Fig. 2 G-L). Of note, Gai2 did not promote β-catenin degradation by this QVL mutant, suggesting a causal relationship between cellular distribution and function (Fig. 4a, b / old Fig. 3 A, B)." I still (respectfully) disagree. This is a correlation because to address causality one should be able to manipulate a physical property (e.g., phase separation) without affecting other chemical properties (e.g., protein-protein association). Since the QVL induces aggregation, we cannot conclude that what drives the biological function is the phase separation because it could be simply aggregation.

3- In response to point 8 ("What is the impact of the Gai2 mutants described in Figure 4 on Axin2 binding and in regulating Axin2 localization?"), the authors did experiments that show that "we did not observe a convincing decrease in binding [of the mutants]". This leaves the authors without support for their model. The mutants should bind less to conductin, or the effects could be due to other alterations caused by the mutations on the G protein.

Reviewer #2 (Remarks to the Author):

The authors have addressed most of my comments on the previous version.

Reviewer #3 (Remarks to the Author):

The authors have addressed my concerns, for a large part satisfactorily. However, a major point of concern that remains unconvincingly addressed is the method for quantification of condensates in cells, while these data present a core element of the presented study.

The authors addressed this point by adding a sentence to the Methods section: "according to the predominant distribution of conductin, cells were classified as cells with diffuse distribution, cells with condensates or cells with membrane recruitment". This method is arbitrary, susceptible to bias, and not in line with current standards in the field. Notably, no mention is made whether scoring was done blind. Moreover, quantification should make use of thresholds to categorize cells (e.g. cells with >5, 5-10, >10 condensates), which is particularly important in cases where differences are small (e.g. Fig 2D,E, sup 10C). Many types of software for image-based particle analysis are available to do this.

Reviewer #1 (Remarks to the Author):

I appreciate the authors' effort to respond to my previous concerns. Many of them have been addressed satisfactorily, but some others remain problematic.

1- Previously I raised the following "to establish definitely the mechanism the authors will need a mutant that specifically disrupt the Gai2-Axin2 interaction." (point 1). In response the authors used a previously described mutant (G184S) that efficiently disrupt binding to RGS GAPs. There are two problems with this: (1) the mutant is not specific for Gai-Axin2, instead it disrupts binding of Gai to many RGS GAPs, so any effect of this mutant could be due to diminished RGS GAP binding, and

We agree with the reviewer that the G184S mutation will also attenuate binding of $G\alpha i2$ to other RGS proteins. Yet, as we found that $G\alpha i2$ directly interacted with the conductin RGS domain, and that G184S attenuated this interaction and induction of conductin polymerization, the most likely interpretation of these findings is that $G\alpha i2$ -conductin binding is required for regulating conductin localization. In support of this model, we identify three additional $G\alpha i2$ mutations that impair conductin RGS binding and, at the same time, induction of polymerization, by revealing that the cancer mutants, which showed reduced induction of conductin polymerization (Supplementary Fig. 10e), interact less with the conductin RGS domain (new Supplementary Fig. 10c, d; please also see point 3 below). These mutations are position-wise unrelated, decreasing the probability for off target effects. Finally, other previous findings such as the perfect correlation between interaction with the conductin RGS domain and induction of conductin polymerization of several $G\alpha$ -proteins, or the required overlap between the $G\alpha i2$ binding site with the conductin RGS aggregon to induce polymerization (QVL mutant) point to the importance of $G\alpha i2$ binding. Thus, collectively, our findings strongly suggest that $G\alpha i2$ induces conductin polymerization via direct binding to its RGS domain.

the effect of this mutation on Axin2 binding is quite modest. Together with their new finding that the interaction is similarly strong with GTPgS and AIF4-loading suggests that Axin2 has an effector-like binding pose rather than GAP-like binding. Thus, the authors should put effort on designing a mutant of Gai2 or Axin2 that specifically disrupt their interaction without affecting other binders.

The G184S mutation attenuated $G\alpha i2$ -conductin interaction highly significantly and by about 50%. Induction of conductin condensates was reduced to a similar degree thereby fitting rather than compromising our model. Structural considerations can explain why the interaction with conductin was disrupted less potently compared to other RGS proteins by this mutation: The introduced serine (blue) has a repellent activity by sterically interfering with the backbone carbonyl (red) of a glutamate in the RGS protein¹. Importantly, in case of conductin, there is more space between the serine (blue) and the respective carbonyl group (cyan), indicating less repulsion (see figure for the reviewer below). As the RGS-like binding model nicely explains the activity of the G184S mutation, we are confident that conductin interacts with $G\alpha i2$ like a typical RGS protein, as already suggested by the crystal structure of the axin RGS domain². Therefore, searching for another mutant that specifically affects $G\alpha i2$ -conductin binding might turn out laborious because of the similar binding mode, and, moreover might not lead to conclusive results because one can never rule out that any other

of the promiscuous interactions of G α i2 or conductin might be affected. We therefore did not follow this advice. We already tried to design mutations in the conductin RGS domain that prevent G α i2 binding based on published structural data. Unfortunately, candidates (e.g. D106A) are close to the silent aggregation site and predicted to alter conductin aggregation (TANGO aggregation algorithm), which will render them non-informative.

The Gly-Ser exchange and the conductin RGS domain (black) were modelled onto the structure of RGS1 (grey) co-crystallized with G α i1 (green) (2GTP). The introduced Ser sidechain (blue) and the backbone of Glu106 in RGS1 (red) and Glu102 in conductin (cyan) are highlighted.

2- In response to my previous point 6 ("the evidence to support a causal relationship between phase separation and function is nonexistent."), the authors state that "We established a conductin mutant (QVL) that did not form condensates upon G α i2 binding (Fig. 3 / old Fig. 2 G-L). Of note, G α i2 did not promote β -catenin degradation by this QVL mutant, suggesting a causal relationship between cellular distribution and function (Fig. 4a, b / old Fig. 3 A, B)." I still (respectfully) disagree. This is a correlation because to address causality one should be able to manipulate a physical property (e.g., phase separation) without affecting other chemical properties (e.g., protein-protein association). Since the QVL induces aggregation, we cannot conclude that what drives the biological function is the phase separation because it could be simply aggregation.

We principally agree with the reviewer that alterations of protein-protein associations include a potential risk to interfere with more than just phase separation in case of any phase separating protein. However, it is the standard in the field to study a functional role of phase separation by mutation of protein regions underlying this process^{3,4,5}, probably because changes of physical parameters may have more unpredictable pleiotropic effects in functional assays.

In our case, we cannot formally rule out that the condensation-inactive conductin QVL mutant shows additional changes in aggregation, which prevents activation by G α i2, as speculated by the reviewer. Therefore, we used a second condensation-inactive conductin mutant that does not rely on induced aggregation but lack of polymerization (conductinM3; Fig. 1d). Importantly, G α i2 did not enhance β -catenin degradation by conductinM3 (new Supplementary Fig. 7a-c). Thus, interference with conductin condensation by two different

molecular mechanisms makes conductin refractory towards activation by G α i2 strongly suggesting a functional role of conductin condensation in β -catenin degradation. Moreover, also from the G α i2 side, mutations that reduced condensate formation of conductin (G184S [Fig. 2d], and cancer mutants K210N, G293E and T328S [Supplementary Fig. 10e]) were less active in inhibiting Wnt/ β -catenin signaling (Supplementary Fig. 7i and Fig. 5h), providing further correlative evidence for a functional role of conductin condensation in Wnt pathway inhibition.

Finally, our proposed mechanism is supported by studies of other groups causally linking β -catenin degradation with polymerization of axin into puncta-shaped condensates, which have recently been shown to form via liquid-liquid phase separation³. Mutational inactivation of the DIX domain⁶, activation of an otherwise silent aggregation site in the axin RGS domain by a cancer mutation⁷ and mutational inactivation of phase separating regions in axin³, all abolished axin condensates and resulted in reduced inhibition of β -catenin. Although some of the provided evidence may also be of correlative nature, we feel that altogether the findings of others and us strongly point to a functional link between condensation and activity of axin proteins.

3- In response to point 8 ("What is the impact of the Gai2 mutants described in Figure 4 on Axin2 binding and in regulating Axin2 localization?"), the authors did experiments that show that "we did not observe a convincing decrease in binding [of the mutants]". This leaves the authors without support for their model. The mutants should bind less to conducting, or the effects could be due to other alterations caused by the mutations on the G protein.

In order to analyze the effect of the cancer mutations on G α i2-conductin interaction more carefully, we changed our assay from pulldowns out of cellular lysates to pulldowns with recombinant proteins using the purified GST-tagged conductin RGS domain and *in vitro* translated G α i2-GFP WT and mutant proteins. We reasoned that any relative reduction in binding will be more robustly identified when starting from a high initial binding, as absolute differences will be increased. Indeed, we found significantly reduced binding for all three cancer mutants compared to WT (new Supplementary Fig. 10c, d), supporting our model that binding of G α i2 regulates conductin distribution and activity.

Reviewer #2 (Remarks to the Author):

The authors have addressed most of my comments on the previous version.

Reviewer #3 (Remarks to the Author):

The authors have addressed my concerns, for a large part satisfactorily. However, a major point of concern that remains unconvincingly addressed is the method for quantification of condensates in cells, while these data present a core element of the presented study. The authors addressed this point by adding a sentence to the Methods section: “according to the predominant distribution of conductin, cells were classified as cells with diffuse distribution, cells with condensates or cells with membrane recruitment”. This method is arbitrary, susceptible to bias, and not in line with current standards in the field. Notably, no mention is made whether scoring was done blind. Moreover, quantification should make use of thresholds to categorize cells (e.g. cells with >5, 5-10, >10 condensates), which is particularly important in cases where differences are small (e.g. Fig 2D,E, sup 10C). Many types of software for image-based particle analysis are available to do this.

We agree with the reviewer that robust quantification of cells with condensates is a prerequisite of our study. We can assure the reviewer that cells with condensates are clearly distinct and readily distinguishable from cells with diffuse conductin distribution, and thus can be reliably quantified via manual counting. To illustrate this, we provide two overview images for the reviewer with indicated diffuse (D) and condensate (C) cells:

Conductin

Conductin + Gxi2

Following the reviewer’s request, we specified the description of the criteria used for classification of the cells in the Methods section. For experiments with smaller differences (e.g. Fig. 2d), quantification was performed in a blinded fashion. We now included a respective sentence in the Methods section, in addition to the statement previously provided in the Reporting Summary.

We appreciate the suggestion of the reviewer to include subcategories for more sensitive analysis of the smaller differences in Fig. 2d and Supplementary Fig. 10c. Indeed, image analysis via the Icy Spot Detector⁸ revealed that induction of condensation by Gxi2 mutants (G184S, K210N, G293E and T328S) was markedly reduced compared to WT when looking at cells with more than 15 condensates (new Supplementary Fig. 4c, d and 10f, g), supporting our previous findings.

Comparison of the old counting method with the new image-based spot detection showed that cells that we classified as “diffuse” displayed 0.6 spots on average, while cells that we classified as “with condensates” after Gxi2 co-expression displayed 24.8 spots on average. The clear difference of image-based detected spots between the two manually grouped categories indicates robustness of the manual categorization method.

1. DiBello PR, *et al.* Selective uncoupling of RGS action by a single point mutation in the G protein alpha-subunit. *J Biol Chem* **273**, 5780-5784 (1998).
2. Spink KE, Polakis P, Weis WI. Structural basis of the Axin-adenomatous polyposis coli interaction. *EMBO J* **19**, 2270-2279 (2000).
3. Nong J, Kang K, Shi Q, Zhu X, Tao Q, Chen YG. Phase separation of Axin organizes the beta-catenin destruction complex. *J Cell Biol* **220**, (2021).
4. Shi B, *et al.* UTX condensation underlies its tumour-suppressive activity. *Nature* **597**, 726-731 (2021).
5. Boija A, *et al.* Transcription Factors Activate Genes through the Phase-Separation Capacity of Their Activation Domains. *Cell* **175**, 1842-1855 e1816 (2018).
6. Fiedler M, Mendoza-Topaz C, Rutherford TJ, Mieszczanek J, Bienz M. Dishevelled interacts with the DIX domain polymerization interface of Axin to interfere with its function in down-regulating beta-catenin. *Proc Natl Acad Sci U S A* **108**, 1937-1942 (2011).
7. Anvarian Z, *et al.* Axin cancer mutants form nanoaggregates to rewire the Wnt signaling network. *Nat Struct Mol Biol*, (2016).
8. Olivo-Marin JC. Extraction of spots in biological images using multiscale products. *Pattern Recogn* **35**, 1989-1996 (2002).

REVIEWERS' COMMENTS

Reviewer #1 (Remarks to the Author):

The authors have improved the manuscript and addressed most of my remaining points. I am sorry to be a stickler for accuracy, but the the authors' response regarding my second point raises exactly the issue that concerns me:

"We principally agree with the reviewer that alterations of protein-protein associations include a potential risk to interfere with more than just phase separation in case of any phase separating protein. However, it is the standard in the field to study a functional role of phase separation by mutation of protein regions underlying this process, probably because changes of physical parameters may have more unpredictable pleiotropic effects in functional assays."

Essentially, I am concerned that what is accepted in the field is not correct and it self-perpetuates paper after paper following a circular logic (it is done because it is accepted in the field, and it is accepted in the field because it is done by many). The fact that there are not good enough approaches for this problem should not justify unsupported conclusions.

I do not have the intention to hinder the publication of this work for this reason, but I think it would be pertinent for the authors to write a paragraph of discussion stating the exact point that we have discussed back and forth, for which they seem to agree that a causative relationship between phase separation and function cannot be unequivocally established by the work presented here and in many other papers due to current limitations of the approaches used in the field. While there is strong correlative evidence presented here, the above mentioned limitations in the field warrant caution and a critical re-evaluation of the approaches established for the work in this area.

Reviewer #3 (Remarks to the Author):

The authors have addressed my remaining concern satisfactorily. The revised manuscript was improved and I have no further issues to report.

Reviewer #1 (Remarks to the Author):

The authors have improved the manuscript and addressed most of my remaining points. I am sorry to be a stickler for accuracy, but the the authors' response regarding my second point raises exactly the issue that concerns me:

"We principally agree with the reviewer that alterations of protein-protein associations include a potential risk to interfere with more than just phase separation in case of any phase separating protein. However, it is the standard in the field to study a functional role of phase separation by mutation of protein regions underlying this process, probably because changes of physical parameters may have more unpredictable pleiotropic effects in functional assays."

Essentially, I am concerned that what is accepted in the field is not correct and it self-perpetuates paper after paper following a circular logic (it is done because it is accepted in the field, and it is accepted in the field because it is done by many). The fact that there are not good enough approaches for this problem should not justify unsupported conclusions.

I do not have the intention to hinder the publication of this work for this reason, but I think it would be pertinent for the authors to write a paragraph of discussion stating the exact point that we have discussed back and forth, for which they seem to agree that a causative relationship between phase separation and function cannot be unequivocally established by the work presented here and in many other papers due to current limitations of the approaches used in the field. While there is strong correlative evidence presented here, the above mentioned limitations in the field warrant caution and a critical re-evaluation of the approaches established for the work in this area.

We agree with the reviewer that we provide strong correlative evidence for our conclusion, which however is not formally unequivocal, if you speculate about unlikely confounding off-target effects. Considering this concern, we toned down interpretations for accuracy (e.g. changing "demonstrating" to "strongly suggesting" in the manuscript on page 8), we mentioned potential off-target effects in the discussion and that they are minimized by employing different mutants, and we included published data functionally linking axin1 condensation and Wnt pathway inhibition, supporting our findings.

At the end, it all comes down to the question of how much evidence you need to be convinced about a finding. Here we show that loss of condensation results in loss of $\text{G}\alpha\text{i}2$ -induced β -catenin degradation for two position-wise unrelated point mutations of conductin that prevent $\text{G}\alpha\text{i}2$ -induced polymerization based on two independent mechanisms (QVL, M3). From the $\text{G}\alpha\text{i}2$ side, we used four point mutants (G184S, K210N, G293E and T328S) that reduced induction of conductin condensation, and at the same time attenuated inhibition of Wnt signaling. With GBZ and AIF_4^- , we used two different cell treatments that promote conductin condensation and enhance conductin-mediated β -catenin degradation, independently of mutations. In addition, we previously showed that an RGS domain mutation induced condensation of conductin and conductin-mediated β -catenin degradation, and that a rescue mutation in the DIX domain that prevented condensation also prevented enhanced β -catenin degradation¹. Moreover, our studies are consistent with strong published data functionally linking condensation of axin1 with Wnt pathway inhibition (please see our answer during the second revision for details)^{2, 3, 4, 5}. Finally, components of the β -catenin destruction complex were shown to be enriched in these condensates, perfectly allowing to understand how polymerization of the scaffold proteins mechanistically promotes β -catenin degradation^{3, 6}.

Thus, there are loss of and gain of function studies on the polymerization/condensation of axin1 and conductin/axin2, in which condensation was altered either via different protein mutations (RGS domain, IDR, DIX domain), co-expression of interactors (APC, G*α*i2) or chemical treatment of cells (AlF₄⁻, GBZ), linking increased condensation with increased inhibition of Wnt/β-catenin signaling. We grant the reviewer that all this evidence may not be unequivocal, if you propose various off-target effects and alternative mechanisms. For us, the data provide sufficient evidence to be thoroughly convinced that condensation of axin proteins is functionally important for β-catenin degradation.

We abstained from discussing a problem to causally link condensation with protein function as limitation in the field because the concern raised by the reviewer is not specific to the condensate field. Following her/his line of argumentation, no mutagenesis-based functional characterization of protein domains would be telling, since a risk for off-target effects remains. An original research article is not the right place to discuss this principal issue.

We thank the reviewer for the fair open statement that she/he does not want to hinder the publication of our work.

Reviewer #3 (Remarks to the Author):

The authors have addressed my remaining concern satisfactorily. The revised manuscript was improved and I have no further issues to report.

We thank the reviewer for the constructive reviewing process.

1. Bernkopf DB, Bruckner M, Hadjihannas MV, Behrens J. An aggregon in conductin/axin2 regulates Wnt/beta-catenin signaling and holds potential for cancer therapy. *Nat Commun* **10**, 4251 (2019).
2. Fiedler M, Mendoza-Topaz C, Rutherford TJ, Mieszczanek J, Bienz M. Dishevelled interacts with the DIX domain polymerization interface of Axin to interfere with its function in down-regulating beta-catenin. *Proc Natl Acad Sci U S A* **108**, 1937-1942 (2011).
3. Nong J, Kang K, Shi Q, Zhu X, Tao Q, Chen YG. Phase separation of Axin organizes the beta-catenin destruction complex. *J Cell Biol* **220**, (2021).
4. Mendoza-Topaz C, Mieszczanek J, Bienz M. The Adenomatous polyposis coli tumour suppressor is essential for Axin complex assembly and function and opposes Axin's interaction with Dishevelled. *Open biology* **1**, 110013 (2011).
5. Anvarian Z, *et al.* Axin cancer mutants form nanoaggregates to rewire the Wnt signaling network. *Nat Struct Mol Biol*, (2016).
6. Thorvaldsen TE, *et al.* Structure, Dynamics, and Functionality of Tankyrase Inhibitor-Induced Degradasomes. *Mol Cancer Res* **13**, 1487-1501 (2015).